# Double-Checker: Enhancing Reasoning of Slow-Thinking LLMs via Self-Critical Fine-Tuning

## Abstract

While slow-thinking large language models (LLMs) exhibit reflection-like reasoning, commonly referred to as the "aha moment", their ability to generate informative critiques and refine prior solutions remains limited. In this paper, we introduce `Double-Checker`, a principled framework designed to enhance the reasoning capabilities of slow-thinking LLMs by fostering explicit self-critique and iterative refinement of their previous solutions. By fine-tuning on our curated 1,730 self-critical instances, `Double-Checker` empowers long-CoT LLMs to iteratively critique and refine their outputs during inference until they evaluate their solutions as correct under self-generated critiques. We validate the efficacy of `Double-Checker` across a comprehensive suite of reasoning benchmarks, demonstrating that iterative self-critique significantly enhances the reasoning capabilities of long-CoT LLMs. Notably, our `Double-Checker` increases the pass@1 performance on challenging AIME benchmarks from 4.4% to 18.2% compared to the original long-CoT LLMs. These results highlight a promising direction for developing more trustworthy and effective LLMs capable of structured self-critique.

## 1 Introduction

Reasoning—the capacity to solve complex tasks by logically connecting facts and drawing conclusions—represents a critical milestone in the quest for human-level AI or Artificial General Intelligence (AGI) [1, 2, 3, 4]. Following the advent of large language models (LLMs), extensive research has sought to further enhance their reasoning ability, spanning more effective pretraining [5, 6], supervised fine-tuning [7, 8, 9, 10, 11], rigorous evaluation [4, 12, 13], and, more recently, reinforcement learning (RL) [14, 15, 16]. In particular, [15] shows that RL with verifiable rewards can push LLMs toward generating *long chains of thought* (long-CoT) [17] and exhibiting reflective-like reasoning behavior, often termed as the "aha moment" [18]. Despite these gains, Recent works [19, 20] suggest that revisiting and refining previous solutions might unlock further improvements, motivating us to integrate the "aha moment" into a systematic "reflect-and-refine" loop.

The key concept lies in the "reflect-and-refine" is *critique*: the model's explicit evaluation of whether a solution is correct and, if needed, how it can be improved [21, 22, 23]. Critique underpins the principle of selectively refining only those solutions that need fixing, thereby preserving originally correct answers [24, 25]. Numerous studies have demonstrated that critique can subsequently be utilized to enhance the quality of generated outputs [22, 23, 26]. For example, [22, 27] train specialized critique-oriented LLMs capable of providing feedback to generator LLMs. However, employing a separate model exclusively for critique introduces additional overhead [22]. Alternatively, [23] proposes integrating critique as a training objective. Nevertheless, the resulting LLMs are unable to leverage self-critique effectively during inference. Furthermore, [19] reports only marginal improvements for Long-CoT LLMs, even after fine-tuning on 100K self-critique examples. These raise an open

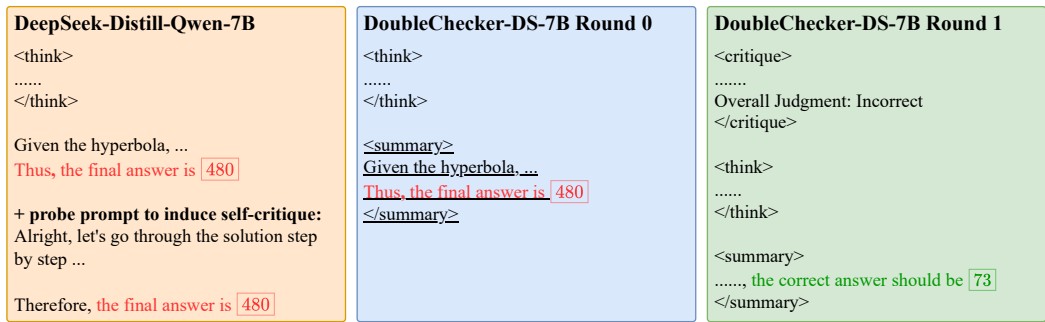

Figure 1: `Double-Checker` correctly solves a math problem in AIME24 leveraging self-critique, while `DeepSeek-Distill-Qwen-7B` still gets the same wrong answer under a self-critical probe.

question: *Do Long-CoT LLMs, which demonstrate reflection-like reasoning, possess the capacity to leverage self-critique to enhance performance? If not, how can we equip them with this ability?*

In this paper, we investigate the integration of reflection-like reasoning with self-critique to enhance the reasoning abilities of slow-thinking LLMs. Specifically, we start by examining whether long-CoT LLMs can leverage self-critique to iteratively refine their prior solutions during inference in a probe-induced manner. Our findings reveal that the occurrence of an "aha moment" does not necessarily indicate the presence of a self-critique mechanism (see Sec. 3.1). For instance, as illustrated in Fig. 1, `DeepSeek-Distill-Qwen-7B` fails to generate informative critiques of its prior solution, ultimately arriving at the same incorrect answer. To address this, we introduce `Double-Checker`, a novel framework designed to empower LLMs to critique and refine their prior solutions iteratively and adaptively. Through a specialized training process that combines direct inference instances with curated critique-refine data (1,730 instances in total), our `Double-Checker` equips long-CoT LLMs with an effective self-critique capability. This enables iterative improvements in performance during inference via self-critique. An example of this process is shown in Fig. 1, where `Double-Checker` successfully resolves a complex math problem using the "reflect-and-refine" approach.

Our main contributions can be summarized as follows:

❶ We investigate the self-critical behavior of long-CoT LLMs via a probing and find that they are unable to generate informative critiques to improve their prior solutions.

❷ We propose `Double-Checker`, a novel framework that pairs direct inference data with carefully curated critique-refine dataset (1,730 in total), enabling LLMs to *iteratively* correct flawed reasoning during inference.

❸ Experiments on a wide range of reasoning benchmarks demonstrate that even with a modest amount of critique data, `Double-Checker` unlocks substantial improvements in accuracy. Notably, our method raises pass@1 performance on challenging AIME benchmarks from 4.4% to 18.2%, underscoring the impact of explicit self-critique.

## 2 Related Work

**Long Chain-of-Thought and Slow Thinking.** The rise of LLMs has driven extensive research aimed at enhancing their reasoning capabilities through a range of strategies. Early efforts include advancements in pretraining methodologies [5, 6], supervised fine-tuning [7, 8, 9, 10, 11], and rigorous evaluation techniques [4, 12, 13]. More recently, RL has emerged as a key paradigm for improving reasoning in LLMs. For instance, [15] demonstrates that RL with verifiable rewards enables models to generate long chains of thought (long-CoT) [17], fostering more structured, multi-step problem-solving skills. This approach has been shown to promote reflective reasoning behaviors, termed as the "aha moments" [18]. These advancements mark significant progress in

LLM reasoning [28]. However, our work reveals a critical limitation: while strong reflection-like reasoning allows LLMs to recognize errors or inconsistencies, it does not inherently ensure robust self-improvement [29, 30]. Additionally, existing self-improvement methods [29, 30] often depend on external tools or explicit feedback mechanisms, making it challenging to guide a single LLM through multiple, reliable rounds of refinement. Addressing these challenges is crucial to unlocking the full potential of self-improvement in LLMs.

**Critique LLMs and Integrated Self-Improvement.** A parallel line of research employs *critique models* or reward estimators to score and refine outputs from a "generator" model, especially in mathematical domains [31, 32, 33, 34]. While effective in principle, this split-architecture strategy requires substantial overhead (running two separate LLMs) or produces numeric feedback that lacks actionable corrections [35]. Other efforts have tried to incorporate critique into a single model's training objective [23] or train on large multi-round self-critique data [19], but with limited gains in *iterative* refinement. Against this backdrop, our work introduces `Double-Checker`, which merges critique and generation into a unified "reflect-and-refine" loop within *one* long-CoT LLM. By carefully curating critique-oriented examples and integrating them with direct-inference data, we equip long-CoT LLMs with the capability to generate meaningful critiques and adaptively refine their prior solutions based on self-generated critiques, ultimately enabling robust self-improvement.

## 3 Method

### 3.1 Aha Moment Does Not Equate to Effective Self-Critique

Previous studies have observed that fast-thinking LLMs often generate uninformative critiques, limiting their capacity for self-improvement [22, 36]. In contrast, slow-thinking LLMs are believed to exhibit self-reflection behaviors, identifying and potentially correcting errors in their reasoning steps [15, 16]. This raises the intriguing question: Can long-CoT LLMs with strong reflection-like reasoning abilities perform effective self-critique? To investigate this, we conduct experiments on AIME24 using `DeepSeek-R1-Distill-Qwen7B` and `DeepSeek-R1-Distill-Qwen32B`, employing a probe to induce self-critique behavior (see Appendix A.1 for detailed settings). We have the following results: ❶ `DeepSeek-R1-Distill-Qwen7B` and `DeepSeek-R1-Distill-Qwen32B` follow the probe prompt and produce informative critiques in only 0% and 8.5% of cases, respectively. ❷ The performance on AIME24 improves slightly after refinement with self-critique (1.6% for `Qwen7B`: 57.1% → 58.7% and 0.8% for `Qwen32B`: 72.1% → 72.9%). These findings suggest that the aha moment does not inherently translate into effective self-critique. While these models demonstrate strong capabilities in reflection-type reasoning, their capacity to autonomously evolve through effective self-critique remains limited.

### 3.2 `Double-Checker` **Framework**

Despite exhibiting the "aha moment," long-CoT LLMs demonstrate limited ability to generate actionable critiques and effectively apply them for iterative self-refinement. We hypothesize that this limitation arises because current long-CoT LLMs are primarily trained for direct inference. As a result, these models do not naturally transition toward interactive refinement through self-critique, even when prompted with carefully designed probes (see Sec. 3.1). To address this gap, we propose `Double-Checker`, a novel framework designed to enable long-CoT LLMs to critique their prior solutions and iteratively refine their reasoning. An overview of `Double-Checker` is depicted in Fig. 2. The following section presents the detailed training and inference process of `Double-Checker`.

#### 3.2.1 Training Process

As shown in Fig. 2 (c), the training process of `Double-Checker` consists of four key steps: 1) Initial Generation, 2) Critique with Answer Correctness, 3) Refinement, 4) Distillation, The first three steps focus on data curation, while the final step involves model training. Start from an original training dataset $\mathcal{D}_{orig} = \{(Q_i, GT_i)\}$, which consists of multiple questions with their corresponding ground-truth answers, we will detail each step below.

**Initial Generation** For each question $Q$, we first obtain its direct inference result from a strong teacher long-CoT LLM $\mathcal{T}$ (e.g., `DeepSeek-R1`) as: $\mathcal{T} : Q \rightarrow T_0 \oplus S_0$, where $S_0$ contains the

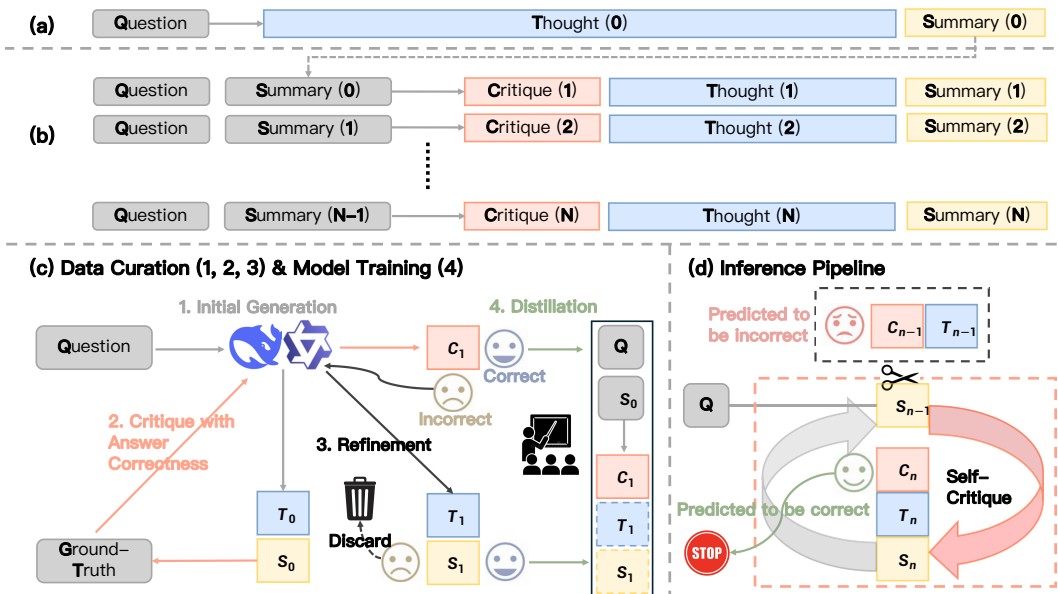

Figure 2: The overview of `Double-Checker`. (a) Direct inference pipeline of long-CoT LLMs: generating a long thought ($T_0$) followed by a summary ($S_0$) that concludes the answer ($A_0$) for the question ($Q$). (b) The inference pipeline of iterative refinement with self-critique. (c) Training stage of our `Double-Checker`. (d) Adaptive inference with self-critique of our `Double-Checker`.

model-generated initial answer $\boxed{A_0}$. The answer $A_0$ is then evaluated for correctness by comparing it with the ground-truth answer $GT$.

**Critique with Answer Correctness**    Given a question $Q$ and its preceding summary $S_0$, we employ a proficient LLM $\mathcal{C}$ to generate detailed critiques. The critique explicitly signals the correctness of the initial answer $A_0$ (correct/incorrect). To optimize critique quality, we employ distinct prompts tailored to correct and incorrect $A_0$ (see App. A.3). This answer correctness signal is indispensable for effective critique generation. Formally, critique generation follows:

$$\mathcal{C} : \{\text{Instruction incorporating Answer Correctness Signal}\} \oplus Q \oplus S_0 \to C_1$$

where $C_1$ adheres to the structured format defined in App. A.2.

**Refinement**    When the initial answer $A_0$ is correct, we collect the corresponding critique $C_1$ and store the triplet $(Q, S_0, C_1)$ into our training set $D_{critique}$. For incorrect $A_0$, we refine the solution using a Refinement long-CoT LLM $\mathcal{R}$, which takes the question $Q$, prior summary $S_0$, and critique $C_1$ as input: $\mathcal{R} : Q \oplus S_0 \oplus C_1 \to T_1 \oplus S_1$, where $T_1$ and $S_1$ represent the refined reasoning and summary, respectively. The refined answer $A_1$ is extracted from $S_1$ and compared to the ground truth $GT$. If $A_1$ matches $GT$, $(Q, S_0, C_1, T_1, S_1)$ is added to $D_{critique}$; otherwise, it is discarded.

**Distillation**    After the data curation stage, training examples are categorized into two formats: $(Q, S_0, C_1)$ for correct answers $A_0$ and $(Q, S_0, C_1, T_1, S_1)$ for incorrect answers. For simplicity, instances with $(Q, S_0, C_1)$ are padded to $(Q, S_0, C_1, T_1', S_1')$ using a predefined template (see App. A.4). To maintain the ability of direct inference of the original long-CoT LLM $\mathcal{M}$, we will mix $D_{critique}$ with a direct inference training set $D_{direc} = \{(Q, T_0, S_0)\}$. Finally, our training set will be $D_{train} = D_{direc} \cup D_{critique}$. The learning objective is

$$\min_{\theta} \left\{ -\frac{1}{|\mathcal{D}_{\text{train}}|} \left( \sum_{\mathcal{D}_{\text{direct}}} \log \mathbb{P}_{\mathcal{M}_\theta}(T_0 \oplus S_0 \mid Q) + \sum_{\mathcal{D}_{\text{critique}}} \log \mathbb{P}_{\mathcal{M}_\theta}(C_1 \oplus T_1 \oplus S_1 \mid Q \oplus S_0) \right) \right\},$$

where $\theta$ is the parameters of long-CoT LLM $\mathcal{M}$ and $|\mathcal{D}_{\text{train}}|$ denotes the number of training examples.

**Algorithm 1** `Double-Checker` Inference Pipeline

---

**Require:** Question $Q$, Long-CoT LLM $\mathcal{M}$, number of iterations $N$
1: Generate initial output $T_0 \oplus S_0 \sim \mathbb{P}_{\mathcal{M}}(\cdot|Q)$                                                  *▷Direct Inference*
2: **for** $n \leftarrow 1$ to $N$ **do**
3:     Critique previous summary and refine $C_n \oplus T_n \oplus S_n \sim \mathbb{P}_{\mathcal{M}}(\cdot|Q \oplus S_{n-1})$    *▷Self-Critique & Refine*
4:     **if** $C_n$ indicates that $A_{n-1}$ (the answer of $S_{n-1}$) is correct **then**        *▷Stopping Criteria*
5:         **return** $S_{n-1}$
6:     **end if**
7: **end for**
8: **return** $S_N$

---

### 3.2.2 Inference Pipeline

We will first introduce a paradigm shift from direct inference (Fig. 2 (a)) to iterative refinement via self-critique (Fig. 2 (b)). Concretely:

- *Round 0 (Direct Inference).* Given a question $Q$, the model $\mathcal{M}$ generates a detailed reasoning chain $T_0$ and a final summary $S_0$, *i.e.*,

$$\mathcal{M} : Q \rightarrow T_0 \oplus S_0.$$

  where $\oplus$ denotes the string concatenation. This baseline (long-CoT) output forms the initial solution.

- *Round 1 (Self-Critique + Refinement).* We now feed both $Q$ and the prior summary $S_0$ to $\mathcal{M}$. The model produces a critique $C_1$ of $S_0$ and then refines the solution into a new thought $T_1$, finally yielding a new summary $S_1$. Formally,

$$\mathcal{M} : Q \oplus S_0 \rightarrow C_1 \oplus T_1 \oplus S_1.$$

- *Round $n$ (Repeated Refinement).* For subsequent rounds ($1 \leq n \leq N$), the model receives $Q \oplus S_{n-1}$, generates $C_n$ to critique the previous summary, and refines the solution into $T_n$ and $S_n$. Symbolically,

$$\mathcal{M} : Q \oplus S_{n-1} \rightarrow C_n \oplus T_n \oplus S_n.$$

  We continue until the model's critique deems the answer correct or a maximum iteration limit $N$ is reached.

**Context Window** The thought $T_i$ is typically lengthy, while the corresponding summary $S_i$ usually encapsulates all the essential information of $T_i$, serving as a concise version of $T_i$. Discarding $T_i$ and retaining only $S_i$ for each refinement round will ensure that the entire refinement process remains within the context window of Long-CoT LLMs.

**Critique Space** The *critique* evaluates the prior summary, assessing whether the answer is correct and proposing actionable suggestions to enhance the solution when needed. Following [22], our critique consists of three components: 1) an analysis of the summary, 2) actionable improvement suggestions, 3) an answer correctness judgment (correct/incorrect). This judgment enables early termination of the iterative refinement process when the solution is deemed correct (see Sec. 3.2). An example of the critique structure is provided in Appendix A.2.

As illustrated in Fig. 2 (d), our `Double-Checker` adopts an iterative refinement pipeline that alternates between: (1) appending the previous summary ($S_{n-1}$) to the input question ($Q$), and (2) generating an informative critique ($C_n$) followed by refining the prior solution ($T_n, S_n$). The process terminates when the critique $C_n$ predicts the correctness of the answer extracted from $S_{n-1}$. The complete inference procedure is formally presented in Algorithm 1. To ensure termination, we define a maximum iteration limit $N$. Although theoretically the process could iterate infinitely until all test examples achieve critique-verified correctness, we impose a finite $N$ in practice to prevent computational divergence due to potential inability to predict correctness for certain cases.

# 4 Experiments and Results

## 4.1 Training Setup

**Training Data Construction**    To construct the original training set $\mathcal{D}_{orig}$, we compile a pool of candidate problems from existing mathematical reasoning datasets: S1.1 [11], DeepMath-103K [37], OpenRS [38], and ORZ-Math-Hard [39]. We filter these candidates using two key criteria: 1) Answer Verifiability: Ensuring ground-truth labels are verifiable via rule-based validation, 2) Difficulty: Selecting problems with appropriate complexity. We get a collection of around 8K high-quality questions, calibrated for both difficulty and correctness. For initial generation, critique annotation, and refinement, we utilize `Qwen3-235B-A22B` [40] and `DeepSeek-R1` [15], i.e., $\mathcal{T} = \mathcal{C} = \mathcal{R}$, but with different instructions. We also incorporate a subset of S1.1 training instances as our $\mathcal{D}_{direct}$, resulting in a total training set $D_{train} = D_{direc} \cup D_{critique}$ of 1,730 training instances. The details of our data sources and filtering process are given in App. B.1.

**Training Details**    We train the Distilled long CoT variants of DeepSeek-R1 (7B and 32B parameters) on our curated training set $\mathcal{D}_{train}$ using full-parameter fine-tuning. The training process employs DeepSpeed ZeRO optimization [41] for efficient memory utilization and FlashAttention2 [42] for accelerated training. Following the implementation in [10], we set the maximum sequence length to 16,384 tokens and adopt a learning rate of $5 \times 10^{-6}$. Implementation details are in App. B.2.

## 4.2 Evaluation Setup

**Evaluation Setting**    We evaluate on AIME24, AIME25, MATH500 [43], and OlympiadBench [44] for mathematical reasoning, and GPQA [12] for multidisciplinary problems. Following [10], we adopt an unbiased pass@1 metric for datasets with only 30 test examples (i.e., AIME24 and AIME25), generating 16 samples with a decoding temperature of 0.6. For the remaining benchmarks, we use greedy decoding. We use vLLM [45] to accelerate inference and set the maximum sequence length to be 32,768 tokens. We set $N$ in Algorithm 1 to 3 for `Double-Checker-DS-7B` and 1 for `Double-Checker-DS-32B`. A brief introduction to different benchmarks and detailed evaluation setting can be found in App. B.3.

**Baselines**    We compare `Double-Checker` against a comprehensive set of baselines, categorized as follows: 1) `DeepSeek-R1-Distill-Qwen` Series (7B, 32B): Strong long-CoT LLMs distilled from `DeepSeek-R1` using 800K examples. 2) `S1.1` (7B, 32B) [11]: Two CoT LLMs distilled from DeepSeek-R1 using 1K high-quality from multiple sources. 3) `LIMO-32B` [10]: A powerful LLM trained on 837 carefully curated examples. 4) `InftyThink` (7B, 32B) [46]: Models trained on 333K examples adapted from OpenR1-Math, with results from the original paper using multi-round interactive inference. 5) `Light-R1` (7B, 32B) [28]: Two-stage SFT (79K data) + RL-trained models. 6) Naive-SFT Baseline (7B, 32B): `DeepSeek-R1-Distill-Qwen` trained on questions of our $\mathcal{D}_{train}$ using standard SFT (without critique learning), which can isolate the contribution of training data. 7) We also include `OpenAI-o1` series [14] and `DeepSeek-R1` for reference. For 4) and 7), we report the results from other papers directly (see App. C.1), and run the remaining baselines by our own.

## 4.3 Main Results

Table 1 summarizes the primary evaluation results on multiple challenging reasoning benchmarks. From Table 1, we have several key observations:

`Double-Checker` **consistently enhances the performance of original long-CoT LLMs across all reasoning benchmarks and model scales.** Our model, `Double-Checker-DS-7B`, outperforms `DeepSeek-Distill-Qwen-7B` by an average of 9.9%, while `Double-Checker-DS-32B` exceeds `DeepSeek-Distill-Qwen-32B` by 11.9%. Notably, `Double-Checker-DS-32B` achieves a significant improvement of 18.2% in pass@1 on the AIME25 benchmark, and `Double-Checker-DS-7B` boosts pass@1 performance on AIME24 by 9.7%, demonstrating the effectiveness of our `Double-Checker` on complex reasoning tasks.

**Self-critique is a crucial factor in driving performance improvements across benchmarks.** Compared to the "naive SFT" baseline, which utilizes the same training problems but excludes explicit critical data, our `Double-Checker` consistently shows superior performance across all

Table 1: Main results (in %) on various benchmarks. The best results within each group are in **bold**. * indicates that the results of the corresponding LLM are sourced from their technical reports or other references due to the cost of using APIs or the unavailability of the LLM. Please refer to Appendix C.1 for corresponding references. The remaining results are from our own runs.

| Model | AIME24 | AIME25 | MATH500 | Olympiad | GPQA | AVG |
|---|---|---|---|---|---|---|
| `OpenAI-o1-Preview*` | 44.6 | 37.9 | 85.5 | 52.1 | 73.3 | - |
| `OpenAI-o1-mini*` | 63.6 | 53.8 | 90.0 | - | 60.0 | - |
| `OpenAI-o1-1217*` | 79.2 | - | 96.4 | - | 75.7 | - |
| `DeepSeek-R1*` | 79.8 | 70.0 | 97.3 | - | 71.5 | - |
| **7B Models** | | | | | | |
| `S1.1-7B` | 17.5 | 19.6 | 76.4 | 37.6 | 19.2 | 34.1 |
| `InftyThink-7B*` | 40.0 | - | 91.7 | - | 51.9 | - |
| `LightR1-7B-DS` | 57.1 | 45.4 | 90.8 | 57.6 | 21.2 | 54.4 |
| `DeepSeek-R1-Distill-7B` (Our base model) | 56.7 | 43.7 | 90.0 | 48.9 | 20.7 | 52.0 |
| `Double-Checker-DS-7B naive SFT` | 57.1 | 43.3 | 88.8 | 56.3 | 24.2 | 53.9 |
| `Double-Checker-DS-7B` | 66.4 | 48.1 | 91.0 | 59.7 | 44.4 | **61.9** |
| **32B Models** | | | | | | |
| `LIMO-32B` | 57.1 | 50.8 | 92.6 | 65.6 | 61.1 | 65.4 |
| `S1.1-32B` | 56.7 | 47.5 | 91.6 | 63.8 | 61.6 | 64.2 |
| `InftyThink-32B*` | 62.5 | - | 96.0 | - | 65.6 | - |
| `QwQ-32B-Preview` | 44.2 | 34.2 | 90.2 | 58.8 | 68.2 | 59.1 |
| `LightR1-32B-DS` | 76.6 | 65.4 | 94.4 | 64.9 | 68.7 | 74.0 |
| `DeepSeek-R1-Distill-32B` (Our base model) | 72.1 | 50.4 | 88.8 | 55.7 | 50.5 | 63.5 |
| `Double-Checker-DS-32B naive SFT` | 74.6 | 60.4 | 93.8 | 65.3 | 64.1 | 71.6 |
| `Double-Checker-DS-32B` | 79.8 | 68.6 | 94.6 | 67.5 | 66.3 | **75.4** |

benchmarks. For instance, on the GPQA benchmark with the 7B model, our `Double-Checker` nearly doubles the performance (44.4% vs. 24.2%). On average, `Double-Checker` outperforms the "naive SFT" baseline by 8% for the 7B model and 3.8% for the 32B model. These results underscore the pivotal role of self-critique in enhancing the reasoning of long-CoT LLMs.

**Our `Double-Checker` demonstrates strong generalizability.** Although the training data primarily consists of math reasoning problems, `Double-Checker` achieves remarkable performance on GPQA, a multidisciplinary QA benchmark. Notably, the only source of reasoning problems from other disciplines is derived from S1.1 [11], and our training problems constitute a proper subset of S1.1. However, our `Double-Checker` even shows significant improvements over S1.1, achieving a 25.2% performance gain for the 7B model and a 4.7% gain for the 32B model on GPQA. Additionally, `Double-Checker`, which relies solely on SFT, performs comparably to models trained with large-scale RL, such as `LightR1`. Specifically, `Double-Checker-7B` outperforms `LightR1-7B-DS` by an average of 7.5%, while `Double-Checker-32B` surpasses `LightR1-32B-DS` by 1.4%. These results align with previous findings that smaller models tend to benefit more from SFT than RL [47, 40].

## 5 Analysis

### 5.1 The Effect of Self-Critique

To verify the effectiveness of self-critique, we evaluate different model configurations on two representative benchmarks: AIME24 (math problem-solving) and GPQA (general QA). Figure 3 displays the results for both 7B and 32B models under the following settings:

- *DS-Distill-Qwen*: A distilled long-CoT baseline.

- *Naive SFT*: Fine-tuned with the same problems as our training set without explicit critical data.

- *N=0,1,2,3*: Our `Double-Checker` approach with $N = 0, 1, 2, 3$ rounds of inference-stage self-critique and refinement.

**Observations and Analysis.** We notice that at the 7B and 32B scales, self-critique leads to immediate accuracy gains over the distilled baseline. In the 7B setting, moving from *N=0* (no refinement) to *N=1* increases performance from 57.3% to 62.7% on AIME24 and from 33.3% to

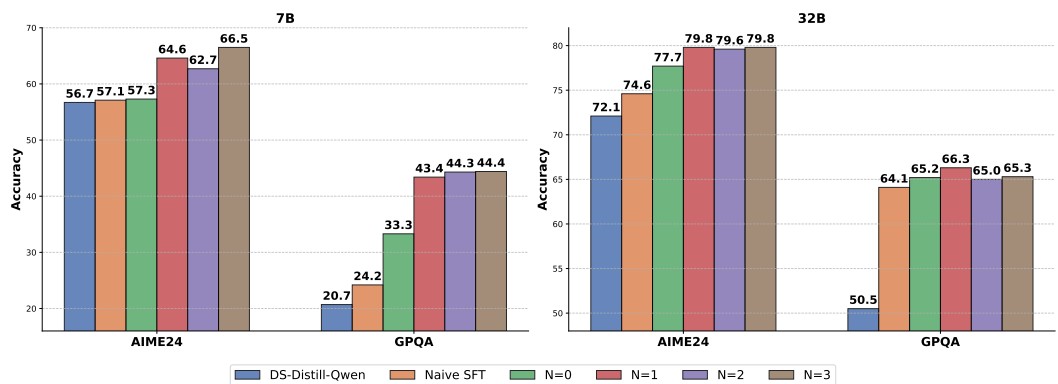

Figure 3: Accuracy comparisons on AIME24 (left) and GPQA (right) for two model sizes (7B and 32B). We compare: (1) `DS-Distill-Qwen` (a distilled baseline), (2) `Naive SFT` (fine-tuning without explicit critique), and (3) `Double-Checker` with varying rounds of self-critique ($N = 0, 1, 2, 3$).

Table 2: Ablation study (in %) on Training Data.

| Model | AIME24 | AIME25 | MATH500 | Olympiad | GPQA | AVG |
|---|---|---|---|---|---|---|
| DeepSeek-R1-Distill-7B | 56.7 | 43.7 | 90.0 | 48.9 | 20.7 | 52.0 |
| Double-Checker-DS-7B naive SFT | 57.1 | 43.3 | 88.8 | 56.3 | 24.2 | 53.9 |
| Double-Checker-DS-7B exclude $\mathcal{D}_{Direct}$ | 57.5 | 44.2 | 86.9 | 55.5 | 19.7 | 52.8 |
| Double-Checker-DS-7B w.o. Qwen3 | 62.0 | 45.6 | 90.8 | 59.5 | 43.9 | 60.4 |
| Double-Checker-DS-7B | **66.4** | **48.1** | **91.0** | **59.7** | **44.4** | **61.9** |

43.4% on GPQA, with further rounds (N=2,3) pushing AIME24 up to 66.5% and GPQA to 44.4%; by contrast, `Naive SFT` yields only modest improvements over `DS-Distill-Qwen`. In the 32B setting, `Double-Checker` starts with a higher performance even at *N=0* (77.7% on AIME24 vs. 72.1%) and achieves 79.8% on AIME24 and 66.3% on GPQA by *N=1*, after which performance saturates (79.6%–79.8% on AIME24; 65.0%–66.3% on GPQA). This suggests that the 32B model acquires self-critique more rapidly than the 7B model. Additionally, incorporating self-critical data during SFT proves beneficial for enhancing direct inference ability as well ($N = 0$ vs. "naive SFT").

These results confirm the strong positive impact of self-critique. Even a single refinement round (N=1) consistently brings notable accuracy gains over baselines, and multiple rounds can yield further improvements—particularly at smaller scales (7B). In contrast, [23] shows even decreased performance of $N = 1$ over $N = 0$ (direct inference). By explicitly learning to critique and update its reasoning, `Double-Checker` effectively bridges the gap between "long chain-of-thought generation" and "iterative self-improvement", resulting in strong reasoning power.

## 5.2 Ablation Study of Training Data

To isolate the contributions of different training data components, we ablate whether (i) we include the original direct-inference data ($\mathcal{D}_{Direct}$), (ii) we use a naive SFT approach without critique data, and (iii) we exclude Qwen3-annotated examples in our curated dataset. Table 2 reports the results on multiple math and reasoning benchmarks.

**Observations and Analysis.** *Naive SFT (`Double-Checker-DS-7B naive SFT`) vs. Baseline.* Simply fine-tuning (without explicit critique data yields a mild improvement over `DeepSeek-R1-Distill-7B` (53.9% vs. 52.0% average). This suggests that fine-tuning on direct inference data (in conventional SFT manner) is beneficial, but the gains are limited in the absence of dedicated critique-and-refine training signals.

*Excluding Direct Inference Data (`exclude` $\mathcal{D}_{Direct}$).* Removing the original direct-inference examples (Round 0 data) degrades average accuracy to 52.8%. This decline underscores the importance of retaining a portion of direct inference data in the training mix to preserve the model's overall reasoning capability. We believe this occurs because training exclusively on $\mathcal{D}_{critique}$ causes the model to lose its ability to perform direct reasoning at $N = 0$.

*Effect of removal of Qwen3 Data (`w.o. Qwen3`).* Removing Qwen3-generated critical examples reduces performance from 61.9% to 60.4% on average. Although not as large a drop as excluding direct data, this shows that Qwen3 training instances further enrich the critique set and boost final performance. We believe that scaling up the critical data could yield further performance gains.

Overall, these results confirm that our curated critique dataset, especially when combined with the original direct-inference examples and auxiliary data from Qwen3, plays a pivotal role in achieving the best performance. In other words, the model benefits from both (i) Conventional long CoT data for maintaining its direct inference ability ($N = 0$) and (ii) explicit self-critique examples for learning to iteratively refine its solutions. (iii) potentially more diverse critical data.

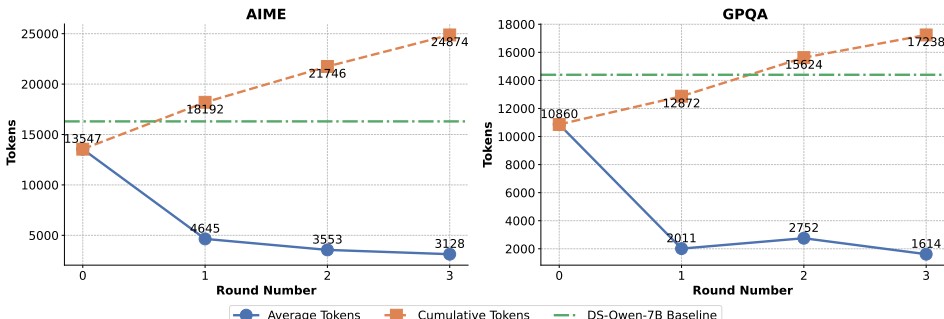

Figure 4: Token usage for AIME24 (left) and GPQA (right). Blue solid line: the per-round average token count, orange dashed line: the cumulative token count over all rounds; green dash-dotted line: the average token consumption for "naive SFT" baseline without iterative refinement.

## 5.3  Analysis of Response Length

The token usage for different rounds of `Double-Checker-DS-7B` for AIME24 and GPQA is shown in Fig. 4. We run up to three refinement rounds ($n = 0, 1, 2, 3$) with `Double-Checker`. We then record both average and cumulative generated tokens used for the full conversation, including thoughts, critiques, and summaries.

**Observations and Analysis.**  On AIME24, the *average tokens per round* drops sharply: from 13.5k at $n = 0$ to 4.6k at $n = 1$, and continues decreasing across subsequent rounds (3.6k at $n = 2$, 3.1k at $n = 3$). A similar pattern holds for GPQA, where the initial 10.9k tokens at $n = 0$ is reduced to roughly 2–3k in the later rounds. Meanwhile, *cumulative tokens* grows steadily with each additional round: for instance, it rises from 13.5k to 24.9k on AIME24 by $n = 3$, and from 10.9k to 17.2k on GPQA. Nevertheless, each new round adds far fewer tokens than the first round. Notably, on GPQA, the total tokens spent in our `Double-Checker` at $N = 2$ is even smaller than "naive SFT".

Interestingly, the tokens spent on the direct inference of our `Double-Checker` is fewer than those of "naive SFT" baseline, indicating that incorporating critical data for SFT could also reduce the token consumption at $N = 0$. While allowing more rounds naturally increases the total token count, each round's contribution is substantially smaller than that of the direct long-CoT baseline. Hence, there is a clear trade-off between improved accuracy (via multiple critique/refine steps) and total token usage. In practice, we find that even a single or two rounds of refinement often suffice to boost correctness without incurring a prohibitive increase in cumulative tokens.

## 6  Conclusion

We have presented `Double-Checker`, a framework that explicitly enforces LLMs to critique and re-fine their previous solutions for self-improvement. Our approach integrates generating reflection-like reasoning and actively correcting potential errors through iterative self-critique. Experimental results on multiple mathematical and multidisciplinary benchmarks demonstrate that `Double-Checker` consistently improves accuracy over comparable baselines, often by large margins. Furthermore, we emphasize the pivotal role of self-critique during inference in enhancing the reasoning capabilities of long-CoT LLMs. `Double-Checker` demonstrates the value of equipping LLMs with a struc-tured critique space and training them to reflect and refine their outputs, facilitating more reliable self-improvement for complex reasoning tasks.

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

# A  Detailed Settings of `Double-Checker`

## A.1  Setting of Meta-Experiments

For the experiment in Sec. 3.1, we use the following probe to induce self-critique ability, which is adapted from [23]. The evaluation setting on AIME24 aligns with our main experiments (see App. B.3).

To evaluate whether the long CoT LLMs can generate informative critiques using the probe, we examine whether their generated content attempts to assess the correctness of their prior solution. If they do not engage in this answer judgment, we consider the critique to be informative.

---

**The Critique Probe**

Please critique each reasoning step of your previous solution to this problem and explain whether your solution is correct or not.

In your critique, you must verify whether any of your previous reasoning steps contain logical or computational errors and suggest ways to correct them if any errors are found.

After critiquing your solution, you must conclude your judgment with 'Conclusion: right/wrong [END]'.

If your conclusion is 'right', stop generating. If your conclusion is 'wrong', improve your previous solution based on your critique, and present the final answer in $\boxed{ANSWER}$.

Critique:

---

Figure 5: The Probe of Our Experiment in Sec. 3.1.

## A.2  The Critique Space

As mentioned in Sec. **??**, our critique consists of three components: 1) an analysis of the summary, 2) actionable improvement suggestions, 3) an answer correctness judgment (correct/incorrect).

We showcase one such example in Fig. 6.

## A.3  Critique Generation

To ensure the quality of critiques, we employ different prompts for correct and incorrect $A_0$ (see Sec. 3.2.1). The prompts are given in Fig. 7 and 8.

## A.4  Distillation Details

After the data curation stage, training examples are categorized into two formats: $(Q, S_0, C_1)$ for correct answers $A_0$ and $(Q, S_0, C_1, T_1, S_1)$ for incorrect answers. For simplicity, instances with $(Q, S_0, C_1)$ are padded to $(Q, S_0, C_1, T_1', S_1')$ using a predefined template (see Fig. 9). Here, $T_1'$ and $S_1'$ are negligible compared to the original reasoning $T_0$ and summary $S_0$. During inference (as described in Alg. 1), if a critique $C_n$ predicts correctness, $T_n$ and $S_n$ can be disregarded since the extracted answer $A_n$ from $S_n$ will match $S_{n-1}$.

# B  Detailed Experimental Setup

## B.1  Training Data

This appendix outlines the filtering criteria applied to our candidate question pool, focusing on two primary principles: difficulty and answer verifiability. We detail the filtering pipeline for each dataset source as follows:

**S1.1 Dataset**  The dataset comprises 1,000 high-quality questions together with their ground-truth answers sourced from multiple domains, with the reasoning trajectory annotated by `DeepSeek-R1`. However, some questions with open-ended answers can not be reliably evaluated using a rule-based

**The Design of Critque Space**

**\<critique\>**
**Analysis**:
The initial approach incorrectly assumes that "exactly two" and "exactly three" categories can be directly summed with those numbers to derive the count of residents owning all four items. This overlooks the distinction between "exactly two" (and "exactly three") and the inclusion of the universal set (candy hearts). The problem requires applying inclusion-exclusion principles to account for overlaps among the four items, not just summing those who own exactly two or three. The solution conflates the total population with direct arithmetic of exclusive categories, leading to an invalid conclusion.

**Improvement suggestions**:
1. Re-examine the definitions of "exactly two" and "exactly three" in the context of a universal set (candy hearts), recognizing that these groups inherently include ownership of the universal item.
2. Use inclusion-exclusion formulas to model overlaps between the four categories systematically.
3. Differentiate between residents owning all four items and those owning subsets (e.g., two specific items plus candy hearts).
4. Set up equations based on known totals (e.g., total residents, counts for each item) to solve for unknown overlaps.

**Overall judgment**:
Incorrect
**\</critique\>**

Figure 6: An Example of the Critique.

Figure 7: The Prompt for Generating Critiques of Incorrect Answers.

evaluation framework. After the removal of these unverifiable instances, we retain 861 questions with both validated answers and corresponding `DeepSeek-R1` annotated reasoning trajectories. We will also use this subset as our $\mathcal{D}_{direc}$.

**DeepMath-103K**  It contains 103K math problems, each annotated with two reasoning paths generated by `DeepSeek-R1`. We retain only 0.6K questions where the `DeepSeek-R1`-annotated reasoning paths are judged to be incorrect compared to ground truth answers by a rule-based evaluation method. We bypass the initial generation stage of `Double-Checker` and instead initialize the reasoning process directly from the `DeepSeek-R1`-annotated reasoning paths.

---
**Prompt for Generating Critiques of Correct Answers**

You are tasked with analyzing your last solution to a problem and providing constructive feedback based on previous solutions. Do NOT provide direct solutions.
**You have already know your last solution to the problem is correct**.
**Important: Do NOT mention something like "you have already know your last solution is correct" in your feedback.**
Structure your response using the following format (without <format> tags):
<format>
Analysis:
{Analysis}

Improvement suggestions:
{Suggestions}

Overall judgment:
{Correct}
</format>

---

Figure 8: The Prompt for Generating Critiques of Correct Answers.

---
**Padding Template**

$T_1'$**:**
<think>
From my last analysis, I have already got the right answer.
</think>

---

$S_1'$:
<summary>
My previous solution is correct. Therefore, the answer is $\boxed{ANSWER}$.
</summary>

---

Figure 9: The Padding Template of Training Examples, where ANSWER is $A_0$.

**OpenRS & ORZ-Math-Hard** The OpenRS dataset contains 7,000 mathematical reasoning problems, and ORZ-Math-Hard comprises 13,000 challenging math problems. Neither dataset provides `DeepSeek-R1`-annotated reasoning trajectories. To filter simple questions, we generate four responses per question using `DeepSeek-R1-Distill-Qwen-7B` with temperature 0.6 and retain only those questions where at least two responses are incorrect. For the remaining questions, we generate four responses using `DeepSeek-R1-Distill-Qwen-32B` with temperature 0.6 and retain only those with at least two incorrect responses. This yields 2.3K difficult problems from OpenRS and 4.4K from ORZ-Math-Hard.

To balance the distribution of correct and incorrect initial summaries ($S_0$), we also exclude correctly solved questions from DeepMath-103K, OpenRS, and ORZ-Math-Hard based on their initial generation by `DeepSeek-R1`.

### B.2 Training Details

We train the on `DeepSeek-R1-Distill-Qwen-7B` and `DeepSeek-R1-Distill-Qwen-32B` our curated training set $\mathcal{D}_{train}$ using full-parameter fine-tuning. The 7B model is trained on either 8×H800 GPUs or 8×H20 GPUs, while the 32B model is trained on either 8×H800 GPUs or 32×H20 GPUs. The training process employs DeepSpeed ZeRO optimization [41] (stage 2 for 7B, and stage 3 for 32B) for efficient memory utilization and FlashAttention2 [42] for accelerated training. Following [10], we set the maximum sequence length to 16,384 tokens and use a batch size of 32, with a learning rate of $5 \times 10^{-6}$. Training times are approximately 0.7 hours per epoch for the 7B model on 8×H20 GPUs and 1.1 hours per epoch for the 32B model on 8×H800 GPUs.

### B.3 Evaluation Details

We evaluate our models on six benchmarks covering diverse reasoning tasks. For mathematical reasoning, we use:

- AIME24/AIME25: Extremely difficult math competition problems, each dataset contains only 30 examples.
- MATH500 [48]: A high school level math problems, the subset of 500 problems is from the original test set of MATH [43].
- OlympiadBench [44]: A benchmark of Olympiad-level problems requiring advanced problem-solving skills. We only include the math subset for our evaluation.

For multidisciplinary reasoning, we use GPQA [12], which covers questions spanning biology, physics, and other domains.

Following [10], we adopt an unbiased pass@1 metric for datasets with limited test examples (AIME24 and AIME25), generating 16 samples with decoding temperature 0.6. For other benchmarks, greedy decoding is used. For baseline models, we use the top-p parameter suggested in their original papers, while for our models, we fix top-p = 1.0. Inference is accelerated using vLLM [45], with a maximum sequence length of 32,768 tokens.

## C   Results Clarification

### C.1   Results Source

As described in Section 4.2, most of the results for open-source LLMs presented in Table 1 are reproduced through our own experiments. However, for LLMs that require API access (e.g., `OpenAI-o1-1217`), we have cited the results from their official technical reports or other sources due to budget constraints. Similarly, the results for InftyThink [46] are sourced from their paper, as their code and models are not publicly accessible.

The sources of these results are detailed as follows:

- Results of InftyThink are from their paper [46].
- The results of `OpenAI-o1-Preview` on AIME24, MATH500 [43], AMC23, GPQA [12], and OlympiadBench-Math [44] are from LIMO [10].
- The results of `OpenAI-o1-mini`, `OpenAI-o1-1217`, `DeepSeek-R1` on AIME24, MATH500, and GPQA are from DeepSeek-R1 report [15].
- The result of `OpenAI-o1-mini` on UGMathBench is from UGMathBench [4].
- The results of `OpenAI-o1-Preview`, `OpenAI-o1-mini`, and `DeepSeek-R1` on AIME25 are from [49].

The evaluation settings of the cited results on MATH500 and GPQA differ from those used in our runs. Specifically, the referenced results on MATH500 and GPQA were obtained using a temperature of 0.6 and the pass@1 metric, which are expected to yield higher performance than the greedy decoding setting in our setting.

## D   Limitation and Broader Impact

**Limitations.**   While `Double-Checker` demonstrates significant gains in long-CoT reasoning, it does rely on a specialized LLMs for initial critical solution generation and on well-defined correctness signals for critique. We have primarily evaluated `Double-Checker` on mathematical problems with clear ground-truth labels; extending it to more open-ended or partially supervised domains remains an interesting direction for future research. Additionally, our current implementation uses a fixed iteration limit $N$ for stopping. Investigating more adaptive or confidence-based stopping criteria could further enhance the robustness and flexibility of our approach. Furthermore, investigating the role of self-critique for reinforcement learning could also be a promising direction.

**Broader Impact.** By incorporating explicit self-critique, `Double-Checker` offers a pathway to more transparent and responsible AI, potentially reducing hallucinations and improving explainability in applications such as education, research, and automated reasoning systems.

