# OpenReview forum: "Double-Checker: Enhancing Reasoning of Slow-Thinking LLMs via Self-Critical Fine-Tuning"
_NeurIPS.cc/2025/Conference — Submitted to NeurIPS 2025_

### Official Review · Reviewer_dezw · 2025-06-13

**Clarity:** 3
**Significance:** 2
**Originality:** 2
**Rating:** 3
**Confidence:** 4

**Summary:**

This paper steers the LLM to do the self-critique-and-refine process to improve the answer correctness. The authors first probe that the current deepseek-distilled models do not self-improve the answer during the long-chain reasoning. Then the authors synthesize the SFT dataset will delicate framework and prompt design in which the critique and refinement processes are included. The SFT training with self-critique makes the performance better in both Math and GPQA.

Overall, I think the main point is clear and the writing is good. However, my biggest concern is whether the innovation part is enough. The self-reflection for LLM reasoning has been extensively studied. The process to synthesize the dataset and train the model with SFT is also not such novel. Forcing LLM to critique and refine the answer is revealed to be effective in case of the testings in the study. However, as authors discussed, whether it will deteriorates model's reasoning capability remains to be answered.

**Questions:**

I am curious whether 8*H800 GPUs can fine-tune 32B models with full parameters, as claimed in Line 836, since H800 only has 80GB memory each. The maximum 16384 tokens and 32 batch size will also consume much memory.

In Line 249, what is the reason that the conclusion of the other study is different from this study.

**Ethical Concerns:**

["NO or VERY MINOR ethics concerns only"]

**Limitations:**

Yes, but in the appendix.

**Paper Formatting Concerns:**

The Appendix part can be put before the declaration checklist for better readability. The limitation section should be moved to the main content.

**Quality:**

3

**Strengths And Weaknesses:**

Strengths: The idea is clear and the writing is easy to follow. The experimental results show the effectiveness of the methd.


My questions are as follows:

1) Lacks the baseline to do both SFT and further RL optimization

2) The authors use Qwen3-235B and DeepSeek-R1 for initial data construction and annotation. That may have the issue of knowledge distillation since the training model 7B and 32B are much smaller. Whether the improvement is not because of the critique-and-refine mechanism but the models are taught to be better with the extra knowledge.

3) The authors only test on GPQA to test generalizability. Whether testing on other domains will be possible?

typo: Line 55, proposeDouble -> propose Double; Line 793, Sec. ??;

---

> ### Author Rebuttal · Authors · 2025-07-31
>
> Dear Reviewer dezw,
>
> We sincerely appreciate the time and effort you have invested in providing constructive feedback to help improve our manuscript. For your concerns, we will address them one by one as follows.
>
> Please note that all results for MATH500, OlympiadBench, and GPQA in this rebuttal have been updated to use a different evaluation setting compared to the original manuscript. These updated settings are designed to better align with the official DeepSeek-R1 report. For further details about these adjustments, please refer to our response to Weakness 1 of Reviewer MqSo.
>
> ## **Weakness 1**
> >Lacks the baseline to do both SFT and further RL optimization
>
> Thank you for your feedback. We would like to clarify that **our DoubleChecker approach primarily relies on SFT, which is why we compare it to a naive SFT baseline.** Additionally, we include LightR1 as a strong comparative baseline since it incorporates both SFT and RL optimization on the same initial model. It is worth emphasizing that LightR1 employs 3k examples for SFT and over 200k examples for GRPO, whereas our DoubleChecker uses significantly fewer resources, utilizing only 1,730 examples for SFT and no RL optimization. Despite this, our method demonstrates competitive and promising results across all baselines.
> We believe that incorporating RL optimization to teach LLMs how to critique could be a valuable direction for future work.
>
> | Model                                | AIME24 | AIME25 | MATH500 | Olympiad | GPQA | LiveCodeBench | AVG   |
> |--------------------------------------|--------|--------|---------|----------|------|---------------|-------|
> | S1.1-7B                              | 17.5   | 19.6   | 80.7    | 42.8     | 41.3 | 12.7  | 35.8 |
> | LightR1-7B-DS                        | 57.1   | 45.4   | 90.3    | 59.0     | 23.1 | 40.9  | 52.6 |
> | DeepSeek-R1-Distill-7B (Our base model)| 56.7 | 43.7   | 92.2    | 59.0     | 35.4 | 38.4  | 54.2 |
> | Double-Checker-DS-7B naive SFT       | 57.1   | 43.3   | 91.4    | 58.6     | 28.7 | 38.5  | 52.9 |
> | Double-Checker-DS-7B                 | 66.4   | 48.1   | 92.7    | 60.0     | 40.4 | 39.4  | 57.8 |
>
> ## **Weakness 2**
> >The authors use Qwen3-235B and DeepSeek-R1 for initial data construction and annotation. That may have the issue of knowledge distillation since the training model 7B and 32B are much smaller. Whether the improvement is not because of the critique-and-refine mechanism but the models are taught to be better with the extra knowledge.
>
> Thank you for this important question. First, we would like to clarify that it is a common practice to generate SFT datasets using larger LLMs [1, 2].
> To disentangle the effects of the critique-and-refine mechanism from the additional knowledge introduced, we have already provided a comparison between our DoubleChecker and the naive SFT baseline in Table 1. From the table, we observe the following:
> + The naive SFT baseline degrades performance compared to the original model (DeepSeek-R1-Distill-7B). Specifically, average performance decreases from 54.2 to 52.9. We hypothesize that this occurs because DeepSeek-R1-Distill-7B has already been distilled from DeepSeek-R1 using approximately 800k training examples, and continuing SFT on a smaller dataset yields little to no improvement.
> + By contrast, our critique-and-refine strategy significantly improves performance, achieving an average score of 57.8. This demonstrates the effectiveness of critique-and-refine for improving the model's performance.
>
> | Model                                | AIME24 | AIME25 | MATH500 | Olympiad | GPQA | LiveCodeBench | AVG   |
> |--------------------------------------|--------|--------|---------|----------|------|---------------|-------|
> | DeepSeek-R1-Distill-7B (Our base model)| 56.7 | 43.7   | 92.2    | 59.0     | 35.4 | 38.4  | 54.2 |
> | Double-Checker-DS-7B naive SFT       | 57.1   | 43.3   | 91.4    | 58.6     | 28.7 | 38.5  | 52.9 |
> | Double-Checker-DS-7B                 | 66.4   | 48.1   | 92.7    | 60.0     | 40.4 | 39.4  | 57.8 |
>
> To further validate our findings, we added an additional experiment where we prompt Qwen3-235B to self-critique. In this experiment, Qwen3-235B was prompted to critique its own outputs after the initial generation. The results in the table below reveal that Qwen3-235B's performance is almost unchanged after self-critique. This demonstrates that **Qwen3-235B does not inherently possess self-critique abilities.**
>
>
> | Model    | AIME24 | AIME25 |
> | -------  | ------ | ------ |
> | Qwen3-235B | 84.5 | 82.3 |
> | +self-critique | 84.3 | 82.7 |
>
> We believe the self-critique ability observed in DoubleChecker arises from the deliberate construction of our critique-and-refine dataset (1.7k examples). As illustrated in Figure 2, the critiques were generated by leveraging the answer correctness signal (True or False), which was determined by comparing the initial generation with the ground-truth answers in the training examples. This approach ensures that the critiques are informative and the judgments are correct.
>
> ## **Weakness 3**
> >The authors only test on GPQA to test generalizability. Whether testing on other domains will be possible?
>
> Thank you for your question. We would like to clarify that research on domain-specific tasks, such as math, code, humanities, and other topics, typically trains and evaluates models on separate datasets tailored to those domains. For example, prior works like DeepMath [3], DeepScaleR [4], and OpenMathReasoning [5] focus solely on mathematical reasoning, while DeepCoder [6] and OpenCodeReasoning [7] focus exclusively on coding tasks. More recently, MegaScience [8] introduced post-training datasets for broader scientific domains.
> However, the methods and datasets developed in these works can be directly adapted to train more general-purpose models, making them highly applicable in industrial settings.
>
> **For this rebuttal, we further evaluate DoubleChecker on LiveCodeBench in addition to GPQA to further test its generalizability.** As shown in the results below, DoubleChecker improves performance on LiveCodeBench even though it was not trained on any coding-specific examples.
> We believe that applying our method to training examples from a more diverse set of general domains would likely enhance its generalizability to unseen domains.
> |Model|GPQA|LiveCodeBench|
> |--------------------------------------|------|---------------|
> |DeepSeek-R1-Distill-7B (Our base model)|35.4|38.4|
> |Double-Checker-DS-7B naive SFT|28.7|38.5|
> |Double-Checker-DS-7B|40.4|39.4|
>
> ## **Question 1**
> >I am curious whether 8*H800 GPUs can fine-tune 32B models with full parameters, as claimed in Line 836, since H800 only has 80GB memory each. The maximum 16384 tokens and 32 batch size will also consume much memory.
>
> Thank you for your question. To address the memory constraints of fine-tuning a 32B model on 8×H800 GPUs, we employed several optimization techniques to reduce memory requirements while maintaining the ability to fine-tune with full parameters. Specifically, we enabled gradient checkpointing, set gradient accumulation to 4, and used DeepSpeed ZeRO-3 [9], which efficiently offloads optimizers, gradients, and parameters to the CPU. These techniques significantly reduce GPU memory usage, allowing us to work within the limitations of 80GB per GPU.
>
> We acknowledge that these optimizations come at the cost of longer training times due to additional communication overhead and checkpointing, but they enable full-parameter training under constraints of computational resources.
>
> ## **Question 2**
> >In Line 249, what is the reason that the conclusion of the other study is different from this study.
>
> In [10], critiques are used solely as an auxiliary task to assist in training large language models (LLMs). The study focuses on teaching models how to generate critiques, but does not explore how to utilize these critiques to improve a model's solutions iteratively. In contrast, our approach not only involves teaching DoubleChecker how to generate critiques but also how to effectively improve prior solutions based on those critiques. Through this critique-and-refine mechanism, our DoubleChecker enables multi-round refinement.
>
>
> Thank you once again for your helpful suggestions! We hope our responses adequately address all your concerns. Should you have any further questions or wish to engage in additional discussion, please feel free to reach out to us.
>
> Sincerely,
> Authors
>
> [1] Deepseek-r1: Incentivizing reasoning capability in llms via reinforcement learning.
>
> [2] Qwen3 technical report.
>
> [3] Deepmath-103k: A large-scale, challenging, decontaminated, and verifiable mathematical dataset for advancing reasoning.
>
> [4] DeepScaleR: Surpassing O1-Preview with a 1.5B Model by Scaling RL
>
> [5] AIMO-2 Winning Solution: Building State-of-the-Art Mathematical Reasoning Models with OpenMathReasoning dataset
>
> [6] DeepCoder: A Fully Open-Source 14B Coder at O3-mini Level
>
> [7] OpenCodeReasoning: Advancing Data Distillation for Competitive Coding
>
> [8] MegaScience: Pushing the Frontiers of Post-Training Datasets for Science Reasoning.
>
> [9] Zero-offload: Democratizing {billion-scale} model training.
>
> [10] Critique fine-tuning: Learning to critique is more effective than learning to imitate.

---

> > ### Author Response · Authors · 2025-08-06
> > **Looking Forward to Further Discussion**
> >
> > Dear Reviewer dezw,
> >
> > We appreciate your valuable feedback and constructive comments! As the discussion deadline approaches, may we kindly ask if our response addresses your concerns? At this stage, we are willing to engage in further discussions with you.
> >
> > Thanks again for your time and effort!
> >
> > Best,
> >
> > Authors

---

> > > ### Comment · Reviewer_dezw · 2025-08-08
> > > **Thanks for your detailed replies, I will remain my current score**
> > >
> > > Thank you for the detailed replies. Overall, I am worried the innovation part may not be sufficient enough as in my initial comment. The self-reflection and fine-tuning have all been well studied. The human prior to introduce self-reflection and correction may limit generalizability across tasks. I will keep my current judgment. Thank you.

---

> ### Author Response · Authors · 2025-08-08
>
> Thank you for your further comment!
>
> ## **About the novelty:**
>
> **Previous works usually focus on prompting [1,2] for fast-thinking LLMs [3]**. Slow-thinking LLMs are believed to have self-reflection during their reasoning. There is no further self-reflection work for these slow-thinking LLMs, which are believed to have reflection-like reasoning.
>
> **Our starting point is that we find "Aha Moment Does Not Equate to Effective Self-Critique" (in Section 3.1), and that is why we use specialized SFT to teach slow-thinking LLMs to self-critique.** A concurrent work [4] pioneers in prompting slow-thinking LLMs to revise previous solutions. Their preliminary results of SFT show no improvements compared with the prompting method, even with 100K training data. Our DoubleChecker falls into the category of improving slow-thinking LLMs by finetuning them to self-critique with strong empirical results.
>
> **Compared with previous works that leverage self-generative feedback, our critique space explicitly gives a final judgment (True/False)** and uses this signal as the stop criterion of the self-improvement loop. We will add a paragraph in "Related Work" to discuss the connection of DoubleChecker to works in the field of "iterative refinement with self-generated feedback".
>
> Additionally, **our SFT data curation pipeline is quite different from traditional SFT, and the effectiveness of our DoubleChecker SFT can be evidenced by strong empirical results compared to the "naive SFT" baseline and several other self-refine baselines** (for detailed settings, please refer to our response to Weakness 4 of Reviewer vDcm):
>
>
>
> | Model | AIME24 | AIME25 | MATH500 | Olympiad | GPQA |
> |-------|--------|--------|---------|----------|------|
> | 7B initial model | 56.7 | 43.7 | 92.2 | 59.0 | 35.4 |
> | +self-refine | 55.0 | 37.5 | 91.6 | 58.4 | 36.8 |
> | +self-critique | 58.7 | 42.9 | 92.3 | 58.7 | 35.5 |
> | +wait | 57.5 | 44.3 | 93.1 | 58.8 | 36.6 |
> | Ours | 66.4 | 48.1 | 92.7 | 60.0 | 40.4 |
>
>
> ## **About generalizability**:
>
> **In our rebuttal to Reviewer vDcm and MqSo, we have added the results of LiveCodeBench. In our discussion with Reviewer vDcm, we have added two social science subjects from MMLU-pro.**
>
> The results are shown below:
>
> | Model                                | GPQA | LiveCodeBench | Business | Law |
> | - | - | - | - | - |
> | DeepSeek-R1-Distill-7B (Our base model)| 35.4 | 38.4  | 61.1 | 14.3 |
> | Double-Checker-DS-7B naive SFT       | 28.7 | 38.5  | 61.7 | 14.9 |
> | Double-Checker-DS-7B                 | 40.4 | 39.4  | 64.3 | 17.2 |
>
> For now, we have included:
> + Four math benchmarks: MATH500, OlypiadBench, AIME24, AIME25
> + One General STEM benchmark: GPQA (including chemistry, biology, physics)
> + One Coding Task: LiveCodeBench
> + Two Social Science Tasks: Business and Law from MMLU-pro
>
> It can be seen that our **Doublechecker improves performance on LiveCodeBench, Business, and Law, even though it was not trained on any coding-specific examples or social science examples**. We believe that applying our method to training examples from a more diverse set of general domains would likely enhance its generalizability to unseen domains. That is to say, **the data curation pipeline and training paradigm of DoubleChecker are directly applicable to other domains**.
>
>
> We hope our further clarification could resolve your concerns.
>
>
>
> [1] Self-Refine. [2] Reflexion. [3] Learning to Check. [4] ThinkTwice

---

> ### Author Response · Authors · 2025-08-09
>
> Reviewer dezw,
>
> We appreciate your further comments.  As the discussion period will end soon, we would like to further explain the novelty:
>
> + **The scope is different**: **Previous works usually focus on prompting （self-refine, reflextion) for fast-thinking LLMs (Learning to Check, S3CMath)**. There is no further self-reflection work for these slow-thinking LLMs, which are believed to have reflection-like reasoning. Our starting point is that we find "Aha Moment Does Not Equate to Effective Self-Critique" (in Section 3.1), and that is why we use specialized SFT to teach slow-thinking LLMs to self-critique.
> + **The specific design is different**: Compared with previous works that leverage self-generative feedback, **our critique space explicitly gives a final judgment (True/False) and uses this signal as the stop criterion of the self-improvement loop**. **Our SFT data curation pipeline is quite different from traditional SFT**, and the effectiveness of our DoubleChecker SFT can be evidenced by strong empirical results compared to the "naive SFT" baseline.
> + **About the effectiveness**: A concurrent work (ThinkTwice) pioneers in prompting slow-thinking LLMs to revise previous solutions. Their preliminary results of SFT show no improvements compared with the prompting method, even with 100K training data. In contrast, our DoubleChecker can substantially improve the performance of original slow-thinking LLMs using only around 1.7K data. Additionally, **we have added a comparison to previous self-refine baselines.** From the results, our DoubleChecker gains better results.
>
> We believe **Novelty is multi-faceted**: **We agree with Reviewer vDcm that our DoubleChecker presents a specific, well-engineered implementation and data formatting strategy tailored for long CoT LLMs.** We believe this contributes both value and novelty to the field. **A well-known example** that underscores this point is the Denoising Diffusion Probabilistic Models (DDPM) introduced in 2020. While the foundational idea originated from the paper “Deep Unsupervised Learning Using Nonequilibrium Thermodynamics” in 2015, it was the DDPM that offered a meticulously engineered approach, making the concept applicable beyond simple toy examples from 2015. We do not intend to claim that our work shines as brightly or represents a milestone akin to DDPM, but this case illustrates that **a well-engineered implementation can also represent a significant contribution in its own right**.
>
> If you feel that we have addressed your concerns, we would greatly appreciate if you would reconsider your score. Thanks again for your time and effort.
>
> Best,
>
> Authors

---

### Official Review · Reviewer_MqSo · 2025-07-01

**Clarity:** 2
**Significance:** 2
**Originality:** 2
**Rating:** 4
**Confidence:** 4

**Summary:**

This paper introduces Double-Checker, a framework designed to enhance the reasoning capabilities of slow-thinking large language models (LLMs) through self-critical fine-tuning. The authors identify that while long-CoT LLMs exhibit "aha moments" in reflection-like reasoning, they lack the ability to generate informative critiques and iteratively refine their solutions. Double-Checker addresses this limitation by fine-tuning models on a curated dataset of 1,730 self-critical instances, enabling iterative self-critique and refinement during inference. The method demonstrates significant improvements across reasoning benchmarks, notably increasing pass@1 performance on AIME benchmarks from 4.4% to 18.2%.

**Questions:**

See the weaknesses part.

**Ethical Concerns:**

["NO or VERY MINOR ethics concerns only"]

**Final Justification:**

The authors' responses have solved my major concerns. I raise my score to 4.

**Limitations:**

See the weaknesses part.

**Quality:**

2

**Strengths And Weaknesses:**

### Strengths

1. Clear Problem Identification: The paper provides a well-motivated investigation showing that "aha moments" in long-CoT LLMs do not translate to effective self-critique capabilities, establishing a clear research gap.

2. Thorough Ablation Studies: The paper includes comprehensive ablation studies examining the contribution of different components (direct data, critique data, number of rounds).

### Weaknesses

1. Inconsistent Baseline Performance and Unclear Evaluation Framework: The reported baseline performance in Table 1 raises significant concerns. While the average performance improvement appears to stem from GPQA results, there are substantial discrepancies with official DeepSeek reports. Specifically, DeepSeek's official documentation reports GPQA accuracy of 49.1% for DeepSeek-R1-Distill-Qwen-7B and 62.1% for the 32B variant, whereas this paper reports only 20.7% and 50.5%, respectively (https://huggingface.co/deepseek-ai/DeepSeek-R1-Distill-Qwen-7B). Similarly, the paper reports 88.8% accuracy for DeepSeek-R1-Distill-32B on MATH500, while the official report shows 94.3%. Critically, the paper fails to specify the evaluation codebase used, which is crucial for mathematical reasoning tasks, as different frameworks employ varying answer extraction methods (e.g., some extract only content within \box{} tags, while others like LightEval are more robust). The results of other baselines reported in the paper do not seem to be as good as the original ones, such as LIMO and s1.1.

2. Limited Domain Generalization: The evaluation is restricted exclusively to mathematical reasoning tasks. A critical question remains unanswered: how does the fine-tuned model perform in other domains? The authors should evaluate performance on coding tasks (e.g., LiveCodeBench) and other reasoning domains to demonstrate the broader applicability of the self-critique mechanism.

3. Incomplete Computational Cost Analysis: Section 5.3's focus solely on token consumption provides a misleading picture of computational overhead. Compared to the DS-Qwen-7B baseline, Double-Checker requires multiple model forward passes for each critique round, which significantly increases inference time. Additionally, Figure 4 should include accuracy curves for each round; if substantial accuracy improvements only occur in later rounds, the claim that "tokens spent on direct inference of Double-Checker are fewer than the naive SFT baseline" becomes meaningless from a practical standpoint.

4. Missing Critical Baseline: The paper lacks a fundamental baseline comparison with "base model + Self-Critique & Refine" without the specialized training, which would help isolate the contribution of the training versus the inference procedure itself.

---

> ### Author Rebuttal · Authors · 2025-07-31
>
> Dear Reviewer MqSo,
>
> We sincerely appreciate the time and effort you have invested in providing constructive feedback to help improve our manuscript. For your concerns, we will address them one by one as follows.
>
> ## **Weakness 1**
> > Inconsistent Baseline Performance and Unclear Evaluation Framework: The reported baseline performance in Table 1 raises significant concerns. While the average performance improvement appears to stem from GPQA results, there are substantial discrepancies with official DeepSeek reports. Specifically, DeepSeek's official documentation reports GPQA accuracy of 49.1% for DeepSeek-R1-Distill-Qwen-7B and 62.1% for the 32B variant, whereas this paper reports only 20.7% and 50.5%, respectively (https://huggingface.co/deepseek-ai/DeepSeek-R1-Distill-Qwen-7B). Similarly, the paper reports 88.8% accuracy for DeepSeek-R1-Distill-32B on MATH500, while the official report shows 94.3%. Critically, the paper fails to specify the evaluation codebase used, which is crucial for mathematical reasoning tasks, as different frameworks employ varying answer extraction methods (e.g., some extract only content within \box{} tags, while others like LightEval are more robust). The results of other baselines reported in the paper do not seem to be as good as the original ones, such as LIMO and s1.1.
>
> Thank you for your insightful comments. We adopted the evaluation framework described in LIMO [1]. Specifically, for test sets with over 100 examples (e.g., MATH500, Olympiad, GPQA), we followed LIMO's approach and used greedy decoding during evaluation to save computational resources. However, upon reviewing the official DeepSeek-R1 report [2], we noted their statement: "Using greedy decoding to evaluate models with long-output reasoning results in higher repetition rates and significant variability across different checkpoints." This limitation likely explains why our results on MATH500, OlympiadBench, and GPQA, especially for some 7B models, are lower than the official DeepSeek-R1 reports.
>
> **To better align the results of the official DeepSeek-R1 and to make the results more robust, we will adopt the same decoding parameters as DeepSeek-R1 report for all test sets (top-p=0.95, temperature=0.6)**. For MATH500, GPQA, and OlympiadBench, we sample 4 times and report the pass@1. **The updated results are as below:**
>
> |Model|AIME24|AIME25|MATH500|Olympiad|GPQA|AVG|
> |--------------------------------------|--------|--------|---------|----------|------|------|
> |7BModels|||||||
> |S1.1-7B|17.5|19.6|80.7|42.8|41.3|40.4|
> |LightR1-7B-DS|57.1|45.4|90.3|59.0|23.1|55.0|
> |DeepSeek-R1-Distill-7B (Our base model) |56.7|43.7|92.2|59.0|35.4|57.4|
> |DeepSeek-R1-Distill-7B+self-critique|58.7|42.9|92.3|58.7|35.5|57.6|
> |DeepSeek-R1-Distill-7B+wait|57.5|44.3|93.1|58.8|36.6|58.1|
> |Double-Checker-DS-7B naive SFT|57.1|43.3|91.4|58.6|28.7|55.8|
> |Double-Checker-DS-7B|66.4|48.1|92.7|60.0|40.4|61.5|
> |32BModels|||||||
> |LIMO-32B|57.1|50.8|93.0|66.1|64.5|66.3|
> |S1.1-32B|56.7|47.5|92.9|58.8|66.8|64.5|
> |QwQ-32B-Preview|44.2|34.2|89.7|63.6|58.8|58.1|
> |LightR1-32B-DS|76.6|65.4|95.2|67.5|68.7|74.7|
> |DeepSeek-R1-Distill-32B (Our base model) |72.1|50.4|93.5|63.0|64.9|68.8|
> |DeepSeek-R1-Distill-32B+self-critique|72.9|54.8|93.5|63.3|64.4|69.8|
> |DeepSeek-R1-Distill-32B+wait|72.8|53.4|94.2|63.8|65.7|70.0|
> |Double-Checker-DS-32B naive SFT|74.6|60.4|93.4|67.2|63.8|71.9|
> |Double-Checker-DS-32B|79.8|68.6|94.3|68.2|66.0|75.4|
>
> Compared with the official reports of DeepSeek-R1, we find that we could mostly reproduce the official results. The only exception is the 7B model on GPQA. However, this is a known inconsistency in the community, as noted in discussions on the Qihoo360/LightR1 GitHub repository, Issue \#53.
>
> |Model|AIME24|MATH500|GPQA|
> |-|-|-|-|
> |Report|72.6|94.3|62.1|
> |Ours|72.1|93.5|64.9|
>
> ## **Weakness 2**
> >Limited Domain Generalization: The evaluation is restricted exclusively to mathematical reasoning tasks. A critical question remains unanswered: how does the fine-tuned model perform in other domains? The authors should evaluate performance on coding tasks (e.g., LiveCodeBench) and other reasoning domains to demonstrate the broader applicability of the self-critique mechanism.
>
> We would like to clarify that research on domain-specific tasks, such as math, code, humanities, and other topics, typically trains and evaluates models on separate datasets tailored to those domains. For example, prior works like DeepMath [3], DeepScaleR [4], and OpenMathReasoning [5] focus solely on mathematical reasoning, while DeepCoder [6] and OpenCodeReasoning [7] focus exclusively on coding tasks. More recently, MegaScience [8] introduced post-training datasets for broader scientific domains.
> However, the methods and datasets developed in these works can be directly adapted to train more general-purpose models, making them highly applicable in industrial settings.
>
> **For this rebuttal, we further evaluate DoubleChecker on LiveCodeBench in addition to GPQA to further test its generalizability.** As shown in the results below, DoubleChecker improves performance on LiveCodeBench even though it was not trained on any coding-specific examples.
> We believe that applying our method to training examples from a more diverse set of general domains would likely enhance its generalizability to unseen domains.
>
> |Model|GPQA|LiveCodeBench|
> |--------------------------------------|------|---------------|
> |DeepSeek-R1-Distill-7B (Our base model)|35.4|38.4|
> |Double-Checker-DS-7B naive SFT|28.7|38.5|
> |Double-Checker-DS-7B|40.4|39.4|
>
> ## **Weakness 3**
> >Incomplete Computational Cost Analysis: Section 5.3's focus solely on token consumption provides a misleading picture of computational overhead. Compared to the DS-Qwen-7B baseline, Double-Checker requires multiple model forward passes for each critique round, which significantly increases inference time. Additionally, Figure 4 should include accuracy curves for each round; if substantial accuracy improvements only occur in later rounds, the claim that "tokens spent on direct inference of Double-Checker are fewer than the naive SFT baseline" becomes meaningless from a practical standpoint.
>
> Inference time is determined by the number of input tokens and output tokens. Due to our design in Section 3.2.2, for each round, the inputs are the question and the previous summary (removing the thinking), which only accounts for a very small tokens compared to generated tokens. For example, for our DoubleChecker-7B on AIME24, the average input tokens are shown in the following table, which is negligible compared to the output tokens.
>
> | | N=0 | N=1 | N=2 | N=3|
> | - | - | - | - | - |
> | avg. input tokens| 450 | 970 | 740 | 700 |
>
> We have included figures of accuracy in Figure 3. After aligning the evaluation setting, the updated results of 7B are shown below. It can be shown that the direct inference of Double-Checker gains comparable performance to naive SFT.
>
> | Model | AIME 24 | GPQA |
> | --    | ---     | ---- |
> | DeepSeek-R1-Distill-7B (Our base model) | 56.7 | 35.4 |
> | Double-Checker-DS-7B naive SFT  | 57.1 | 28.7 |
> | Double-Checker-DS-7B N = 0 | 57.3 | 28.5 |
> | Double-Checker-DS-7B N = 1 | 64.6 | 40.1 |
> | Double-Checker-DS-7B N = 2 | 62.7 | 40.1 |
> | Double-Checker-DS-7B N = 3 | 66.5 | 40.4 |
>
> ## **Weakness 4**
> >Missing Critical Baseline: The paper lacks a fundamental baseline comparison with "base model + Self-Critique & Refine" without the specialized training, which would help isolate the contribution of the training versus the inference procedure itself.
>
> Thank you for your suggestion! **As suggested, we add the baseline "base model + Self-Critique " in the following table.**
> From the results, the majority of the performance improvement of DoubleChecker stems from the specialized training process rather than the inference procedure itself.
>
> |Model|AIME24|AIME25|MATH500|Olympiad|GPQA|
> |--------------------------------------|--------|--------|---------|----------|------|
> | 7B Models
> |DeepSeek-R1-Distill-7B (Our base model) |56.7|43.7|92.2|59.0|35.4|
> |DeepSeek-R1-Distill-7B+self-critique|58.7|42.9|92.3|58.7|35.5|
> |Double-Checker-DS-7B naive SFT|57.1|43.3|91.4|58.6|28.7|
> |Double-Checker-DS-7B|66.4|48.1|92.7|60.0|40.4|
> | 32B Models|
> |DeepSeek-R1-Distill-32B (Our base model)|72.1|50.4|93.5|63.0|64.9|
> |DeepSeek-R1-Distill-32B+self-critique|72.9|54.8|93.5|63.3|64.4|
> |Double-Checker-DS-32B naive SFT|74.6|60.4|93.4|67.2|63.8|
> |Double-Checker-DS-32B|79.8|68.6|94.3|68.2|66.0|
>
>
>
> Thank you once again for your helpful suggestions! We hope our responses adequately address all your concerns. Should you have any further questions or wish to engage in additional discussion, please feel free to reach out to us.
>
> Sincerely,
>
> Authors
>
> [1] Limo: Less is more for reasoning.
>
> [2] Deepseek-r1: Incentivizing reasoning capability in llms via reinforcement learning.
>
> [3] Deepmath-103k: A large-scale, challenging, decontaminated, and verifiable mathematical dataset for advancing reasoning.
>
> [4] DeepScaleR: Surpassing O1-Preview with a 1.5B Model by Scaling RL
>
> [5] AIMO-2 Winning Solution: Building State-of-the-Art Mathematical Reasoning Models with OpenMathReasoning dataset
>
> [6] DeepCoder: A Fully Open-Source 14B Coder at O3-mini Level
>
> [7] OpenCodeReasoning: Advancing Data Distillation for Competitive Coding
>
> [8] MegaScience: Pushing the Frontiers of Post-Training Datasets for Science Reasoning.

---

> > ### Author Response · Authors · 2025-08-06
> > **Looking Forward to Further Discussion**
> >
> > Dear Reviewer MqSo,
> >
> > We appreciate your valuable feedback and constructive comments! As the discussion deadline approaches, may we kindly ask if our response addresses your concerns? At this stage, we are willing to engage in further discussions with you.
> >
> > Thanks again for your time and effort!
> >
> > Best,
> >
> > Authors

---

> > ### Comment · Reviewer_MqSo · 2025-08-08
> >
> > Thanks for the detailed rebuttal.
> >
> > I still have some questions about the response to "Incomplete Computational Cost Analysis". It seems that simply counting the input tokens of the problem does not equate to the computational cost, because the model needs to perform multiple inferences, and the model's previous output will also serve as the input for subsequent inferences. Therefore, compared with the baseline, it would be more appropriate to give the total inference time.

---

> ### Author Response · Authors · 2025-08-08
>
> Thank you for your further feedback!
>
> Regarding the "Incomplete Computational Cost Analysis":
>
> First, we would like to clarify that the table below displays **the total number of input tokens (including the problem and previous summary)** for each round.
>
> | | N=0 | N=1 | N=2 | N=3|
> | - | - | - | - | - |
> | avg. input tokens| 450 | 970 | 740 | 700 |
> | input structure| $Q$ | $Q + S_0$| $Q+S_1$| $Q+S_2$|
>
> **Due to our special design in Section 3.2.2, we discard the content in the thinking process and only include the summary as the input for the next round**. That is to say,
> + For the $N=0$, the input is the question $Q$ itself, the output is $T_0 + S_0$, where $T_0$ is the reasoning content, $S_0$ is the summary;
> + For $N=1$, we only include $Q+S_0$ as our input, the output is $C_1 + T_1 + S_1$, where $C_1$ is the critique;
> + For $N=2$, we only include $Q+S_1$ as our input; the output is $C_2 + T_2 + S_2$, and so on.
>
> **As shown in the above table, the input tokens are negligible compared to the output tokens.**
>
> **As suggested, we also include an analysis of total inference time for DoubleChecker on AIME (per response) in the table below.** The inference time per response for the **"Naive SFT baseline" is 9.42 seconds**. Note that the following inference time is reported using one H20 GPU.
>
> | | N=0 | N=1 | N=2 | N=3|
> | - | - | - | - | - |
> | avg time for this round| 7.78 | 1.69 | 1.49 | 0.95 |
> | avg cummulative time| 7.78 | 9.47 | 10.96 | 11.92 |
>
> We hope our reply could resolve your concern.
>
> Best,
>
> The Authors

---

> > ### Comment · Reviewer_MqSo · 2025-08-09
> >
> > Thank you for the detailed response. I will raise my score.

---

> > > ### Author Response · Authors · 2025-08-09
> > >
> > > We sincerely appreciate your thoughtful suggestions to improve the quality of our work! We are glad that your concerns have been resolved, and we truly appreciate your decision to raise the score! We will make sure to update the corresponding results and analysis as you suggested.

---

### Official Review · Reviewer_vDcm · 2025-07-02

**Clarity:** 2
**Significance:** 2
**Originality:** 2
**Rating:** 4
**Confidence:** 4

**Summary:**

This paper introduces so-called Double-Checker, a framework aimed at improving the reasoning of "slow-thinking" or long-CoT. The authors first observe that existing long-CoT models, despite showing signs of reflection, struggle to effectively critique and refine their own incorrect solutions. The proposed solution involves fine-tuning these models on a curated dataset of 1,730 instances that explicitly teach a self-critique and refinement loop. During inference, the fine-tuned model iteratively generates a solution, critiques it, and refines it until the internal critique judges the answer to be correct. The authors demonstrate that this method significantly boosts performance on several reasoning benchmarks, most notably improving pass@1 accuracy on the challenging AIME benchmarks.

**Questions:**

-	The data curation pipeline relies heavily on powerful teacher models (Qwen3-235B, DeepSeek-R1). To what degree are the observed performance gains attributable to knowledge distillation from these models versus the learning of a genuine, generalizable self-correction skill? What would happen if the teacher models used for data generation were the same size and capability as the student models being trained?

-	Given that your framework is iterative a test time which requires sampling multiple rounds from the llm, How does your framework compare in performance to other test-time scaling strategies?

-	I am not sure I agree that “critique models or reward estimators” are a parallel line of research. It seems like your method is very similar except your original large LLM is being used as the critique model. In this vein, how does your method compare to other critique methods which use critique models? What is the performance gap and compute trade-off compared to these methods? Keep in mind that while the split architecture strategy requires two LLMs the second one can be much smaller than the first and that would more than offset the additional overhead.

-	Does your improved performance carry over to other testing areas? I believe this is needed if you want to claim “strong generalizability”.

-	In Section 3.2.1 where the authors generate fine-tuning data using a dedicated refinement model to improve the quality of the initial critique content, there is no explicit explanation whether this operation will introduce reasoning traces that are unnecessary to yield the final result.

-	When comparing the results of the Double-Checker framework with other baselines in Table 1 in Section 4.3, consider including comparison between the fine-tuning approach with other self-correction / self-critique framework(s) that is either prompt-based (i.e. with no fine-tuning involved) or fine-tuning based (e.g. Learning to Check: Unleashing Potentials for Self-Correction in Large Language Models URL: https://arxiv.org/abs/2402.13035). This could make the result more comprehensive and demonstrate more clearly the advantage of fine-tuning in LLM’s ability to conduct self-critique.

**Ethical Concerns:**

["NO or VERY MINOR ethics concerns only"]

**Final Justification:**

I have read the response from the authors. One of the key weaknesses, the narrow scope of evaluation and generalizability, is still a concern, as shared with other reviews. While the rebuttal has provided results of DoubleChecker on LiveCodeBench, the paper could benefit from a more comprehensive evaluation. My other questions were answered by the response. I increased my score to 4, but I would not argue against it if the paper is rejected.

**Limitations:**

Yes

**Quality:**

2

**Strengths And Weaknesses:**

Strength:

-	The paper's main strength is its impressive empirical results. The Double-Checker framework achieves substantial performance gains over strong long-CoT baselines across multiple challenging benchmarks, including AIME, MATH500, and GPQA. The 18.2% absolute improvement on AIME25 for the 32B model is particularly noteworthy.

-	The authors demonstrate that significant improvements can be achieved by fine-tuning on a relatively small dataset of only 1,730 high-quality, curated instances. This highlights the value of data quality over quantity for teaching complex reasoning skills.

Weakness:

-	The core concept of iterative refinement with self-generated feedback is not new. The framework is conceptually similar to prior work like Self-Refine and Reflexion. The main contribution of Double-Checker appears to be a specific, well-engineered implementation and data formatting strategy for long-CoT models, rather than a significantly novel idea. The paper essentially boils down to finetuning on self-critique data and accuracy may be inflated by simply training on more domain-similar data (shown by improvements in the naïve-SFT baseline)

-	Domain of testing is narrow, mainly limited to math and logic datasets

-	The data curation pipeline is a critical component, yet it relies on extremely powerful models like Qwen3-235B and DeepSeek-R1 as "teachers" to generate the initial solutions, critiques, and refinements. This raises a significant concern: the performance gains may stem from distilling the superior capabilities of these teacher models into the smaller student models, rather than from teaching a generalizable skill of self-critique. This dependency limits the method's accessibility and scalability, as it requires access to SOTA models for data creation.

-	The comparison being drawn is between an iterative double-checking framework and other reasoning model baselines. I am not convinced this is a fair comparison and that other self-reflection baselines or at least multiple sampling strategies /test-time scaling strategies would be a more accurate comparison.

-	The Double-Checker N=0 model (trained on critique data but performing only direct inference) consistently outperforms the naive SFT baseline. This implies that the data format itself, not just the iterative inference, is a major source of improvement. Therefore, framing the main results as a comparison between the iterative Double-Checker (N>0) and the naive SFT model conflates the benefits of the data format with the benefits of the iterative process, potentially overstating the impact of the latter. Also, additional baselines comparation between Double-Checker and other similar approaches would greatly enhance the paper.

---

> ### Author Rebuttal · Authors · 2025-07-31
>
> Dear Reviewer vDcm,
>
> We sincerely appreciate the time and effort you have invested in providing constructive feedback to help improve our manuscript. For your concerns, we will address them one by one as follows.
>
> Please note that all results for MATH500, OlympiadBench, and GPQA in this rebuttal have been updated to use a different evaluation setting compared to the original manuscript. These updated settings are designed to better align with the DeepSeek-R1 report. For details, please refer to our response to Weakness 1 of Reviewer MqSo.
>
> ## **Weakness 1**
> >Iterative refinement with self-generated feedback is conceptually similar to prior work like Self-Refine and Reflexion. Accuracy may be inflated by simply training on more domain-similar data (shown by improvements in the naive-SFT)
>
> Previous works usually focus on prompting [1,2] for fast-thinking LLMs [3]. Slow-thinking LLMs are believed to have self-reflection during their reasoning. A concurrent work [4] pioneers in prompting slow-thinking LLMs to revise previous solutions. Their preliminary results of SFT show no improvements compared with the prompting method, even with 100K training data.
> Our DoubleChecker falls into the category of improving slow-thinking LLMs by finetuning them to self-critique with strong empirical results.
> Compared with previous works that leverage self-generative feedback, our critique space explicitly gives a final judgment (True/False) and uses this signal as the stop criterion of the self-improvement loop.
> We will add a paragraph in "Related Work" to discuss the connection of DoubleChecker to works in the field of "iterative refinement with self-generated feedback".
>
> For concerns about the results of the naive SFT baseline, please refer to our response to weakness 5 and our reply to weakness 1 of Reviewer MqSo.
>
> ## **Weakness 2**
> >Domain of testing is narrow
>
> Research on domain-specific tasks, such as math, code, humanities, and other topics, typically trains and evaluates models on separate datasets tailored to those domains. For example, prior works like DeepMath, DeepScaleR, and OpenMathReasoning focus solely on mathematical reasoning, while DeepCoder and OpenCodeReasoning focus exclusively on coding tasks. More recently, MegaScience introduced post-training datasets for broader scientific domains.
> However, the methods and datasets developed in these works can be directly adapted to train more general-purpose models, making them highly applicable in industrial settings.
>
> **For this rebuttal, we further evaluate DoubleChecker on LiveCodeBench to test its generalizability.** As shown in the results below, DoubleChecker improves performance on LiveCodeBench even though it was not trained on any coding-specific examples.
> We believe that applying our method to training examples from a more diverse set of general domains would likely enhance its generalizability to unseen domains.
>
> |Model|LiveCodeBench|
> |-|-|
> |7B initial model|38.4|
> |naive SFT|38.5|
> |ours|39.4|
>
> ## **Weakness 3**
> >The data curation pipeline relies on Qwen3-235B and DeepSeek-R1. The performance gains may stem from distilling the superior capabilities of these models into the smaller models, rather than from self-critique.
>
> First, it is a common practice to generate SFT datasets using larger LLMs [5,6,7,8].
> From the following table, we have:
> + The naive SFT baseline degrades performance compared to the original model. We hypothesize that this occurs because DeepSeek-R1-Distill-7B has already been distilled from DeepSeek-R1 using approximately 800k training examples, and continuing SFT on a smaller dataset yields little to no improvement.
> + By contrast, our critique-and-refine strategy significantly improves performance. This demonstrates the effectiveness of critique-and-refine for improving the model's performance.
>
> |Model|AIME24|AIME25|MATH500|Olympiad|GPQA|LiveCodeBench|AVG|
> |-|-|-|-|-|-|-|-|
> |7B initial model|56.7|43.7|92.2|59.0|35.4|38.4|54.2|
> |naive SFT|57.1|43.3|91.4|58.6|28.7|38.5|52.9|
> |DC-7B|66.4|48.1|92.7|60.0|40.4|39.4|57.8|
>
> To further validate our findings, we added an additional experiment, where we prompt Qwen3-235B to self-critique. The results in the table demonstrates that **Qwen3-235B does not inherently possess self-critique abilities.**
> |Model|AIME24|AIME25|
> |-|-|-|
> |Qwen3-235B|84.5|82.3|
> |+self-critique|84.3|82.7|
> ## **Weakness 4**
> >At least multiple sampling strategies /test-time scaling strategies would be a more accurate comparison.
>
> To address your concerns, **we introduce three additional baselines.** For clarity, we follow the same notation as in Section 3.2.2:
> - Baseline 1: DS-7B + self-refine
>   We adopt a similar approach to GSM8K "Self-Refine" [1]. After the initial generation, the model is prompted with:
>   There is an error in the solution above. To find the error, go through the semantically complete solution and check if everything looks good."
>   The test input takes the form: $Q + T_0 + S_0 + P$, where $P$ is the self-refine prompt.
> - Baseline 2: DS-7B + self-critique
>   This baseline evaluates the model’s performance in a self-critique setting [4] without any fine-tuning. After generating an initial solution, we concatenate the previous summary to the question and prompt the model for self-critique. That is, the input is: $Q + S_0 + P$, where $P$ is the self-critique prompt.
> - Baseline 3: DS-7B  + wait
>   We also introduce a sequential test-time scaling approach by appending the word "wait" before the `</think>` token in the prompt [8]. The input is $Q + T_0 + wait$.
>
> |Model|AIME24|AIME25|MATH500|Olympiad|GPQA|
> |-|-|-|-|-|-|
> |7B initial model|56.7|43.7|92.2|59.0|35.4|
> |+self-refine|55.0|37.5|91.6|58.4|36.8|
> |+self-critique|58.7|42.9|92.3|58.7|35.5|
> |+wait|57.5|44.3|93.1|58.8|36.6|
> |Ours|66.4|48.1|92.7|60.0|40.4|
>
> From the results, we observe:
> + Baseline 1 decreases performance across most tasks. Upon inspecting the outputs, we find that the model does not generate any thinking tokens (new tokens within </think>): the output either repeats the input prompt $P$ or makes superficial corrections without meaningful reflection.
> + While baseline 2 and 3 improve performance slightly over the initial model, our DoubleChecker gains the best results.
>
> ## **Weakness 5**
> >The N=0 outperforms the naive SFT baseline. This implies that the data format is a major source of improvement.
>
> Thank you for your comments! After aligning the evaluation setting with the DeepSeek-R1 (see our reply to weakness 1 of Reviewer MqSo), the results are shown below.
> |Model|AIME24|GPQA|
> |-|-|-|
> |7B initial model|56.7|35.4|
> |naive SFT|57.1|28.7|
> |Ours N=0|57.3|28.5|
> |Ours N=1|64.6|40.1|
> |Ours N=2|62.7|40.1|
> |Ours N=3|66.5|40.4|
> |-|-|-|
> |32B initial model|72.1|64.9|
> |naive SFT|74.6|63.8|
> |Ours N=0|77.7|64.0|
> |Ours N=1|79.8|66.0|
>
> This suggests that the major source of improvement is the iterative self-critique ability (see also our reply to weakness 3).
>
> ## **Question 1**
> >To what degree are the performance gains attributable to knowledge distillation from these models versus the learning of a generalizable self-correction skill? What would happen if the teacher models used for data generation were the same size and capability as the student models being trained?
>
> For the first question, we kindly refer you to our response to Weakness 3.
> Regarding the second question, we have tried to collect self-critique data from small models and found that smaller models tend to produce excessively long incorrect responses. Moreover, many of these incorrect responses fail to include the </think> delimiter in their generations, which is crucial for explicitly identifying and summarizing the incorrect solution.
>
> ## **Question 2**
> > How does your framework compare in performance to other test-time scaling strategies?
>
> Please see our response to weakness 4.
>
> ## **Question 3**
> >I am not sure I agree that “critique models or reward estimators” are a parallel line of research.
>
> Previous critique models are reward estimators that output a numerical signal for the whole solution (outcome reward model, ORM) or for each reasoning step (process reward model, PRM). However, critique in our paper means self-generative verbal feedback in natural language, which is quite different from these reward estimators. These ORMs or PRMs are typically used to do RL training [9,10] or data selection [11].
>
> ## **Question 4**
> >Results of other testing areas?
>
> Please refer to our reply for Weakness 2.
> ## **Question 5**
> >The authors generate fine-tuning data using a refinement model to improve the quality of the initial critique content, whether this will introduce reasoning traces that are unnecessary to yield the final result.
>
> As described in L128-129, during the refinement stage, we explicitly discard reasoning traces that lead to incorrect final results. However, we acknowledge that even amongst the reasoning traces that yield correct answers, there is potential for overthinking, where the model may generate overly verbose or unnecessarily intricate reasoning paths that are not strictly essential for deriving the final result.
> Addressing the issue of overthinking is still an open problem. We appreciate your suggestion and recognize that mitigating overthinking could further enhance the efficiency and interpretability of the refined reasoning content. We will consider this in our future work to improve data refinement methodologies.
>
> ## **Question 6**
> >Consider including other self-correction frameworks.
>
> Please refer to our response to Weakness 4 and Weakness 1.
>
> Thank you once again for your helpful suggestions! We hope our responses adequately address all your concerns. Should you have any further questions or wish to engage in additional discussion, please feel free to reach out to us.
>
> Sincerely,
>
> Authors
>
> [1] Self-Refine. [2] Reflexion. [3] Learning to Check. [4] ThinkTwice. [5] DeepSeek-R1. [6] Qwen-3. [7] LIMO. [8] S1. [9] Let's Verify Step by Step. [10] Math-Shepherd. [11] Qwen2.5Math.

---

> > ### Author Response · Authors · 2025-08-06
> > **Looking Forward to Further Discussion**
> >
> > Dear Reviewer vDcm,
> >
> > We appreciate your valuable feedback and constructive comments! As the discussion deadline approaches, may we kindly ask if our response addresses your concerns? At this stage, we are willing to engage in further discussions with you.
> >
> > Thanks again for your time and effort!
> >
> > Best,
> >
> > Authors

---

> ### Comment · Reviewer_vDcm · 2025-08-07
>
> I have read the response from the authors. One of the key weaknesses, the narrow scope of evaluation and generalizability, is still a concern, as shared with other reviews. While the rebuttal has provided results of DoubleChecker on LiveCodeBench, the paper would benefit from a more comprehensive evaluation.

---

> ### Author Response · Authors · 2025-08-07
>
> Dear Reviewer vDcm,
>
> Thank you for your further helpful feedback, and we truly appreciate your decision to raise the score!
>
> About the concern of evaluation and generalizability, we **further added two social science subjects from MMLU-pro**.
>
> The results are shown below:
>
> | Model                                | GPQA | LiveCodeBench | Business | Law |
> | - | - | - | - | - |
> | DeepSeek-R1-Distill-7B (Our base model)| 35.4 | 38.4  | 61.1 | 14.3 |
> | Double-Checker-DS-7B naive SFT       | 28.7 | 38.5  | 61.7 | 14.9 |
> | Double-Checker-DS-7B                 | 40.4 | 39.4  | 64.3 | 17.2 |
>
> For now, we have included:
> + Four math benchmarks: MATH500, OlypiadBench, AIME24, AIME25
> + One General STEM benchmark: GPQA (including chemistry, biology, physics)
> + One Coding Task: LiveCodeBench
> + Two Social Science Tasks: Business and Law from MMLU-pro
>
> It can be seen that our **Doublechecker improves performance on LiveCodeBench, Business, and Law, even though it was not trained on any coding-specific examples or social science examples**. We believe that applying our method to training examples from a more diverse set of general domains would likely enhance its generalizability to unseen domains. That is to say, **the data curation pipeline and training paradigm of DoubleChecker are directly applicable to other domains**.
>
> Additionally, we would like to clarify that research on domain-specific tasks, such as math, code, humanities, and other topics, typically trains and evaluates models on separate datasets tailored to those domains. For example, prior works like DeepMath, DeepScaleR, and OpenMathReasoning focus solely on mathematical reasoning, while DeepCoder and OpenCodeReasoning focus exclusively on coding tasks. More recently, MegaScience introduced post-training datasets for broader scientific domains. However, the methods and datasets developed in these works can be directly adapted to train more general-purpose models, making them highly applicable in industrial settings.
>
> In our opinion, the training process of current LLMs usually incorporates data from all possible domains to make it more "general" to apply to possible unseen domains. As evidenced by our results above, our Doublechecker is trained on math exclusively, and could in some sense, generalize to coding and social science. If more data from other domains could be included during training, we believe our DoubleChecker could be more generalizable. Due to computational and time limitations, we could only show the results of training on math examples, which is a typical way people from academia do (e.g., DartMath, DeepScaleR, OpenMathReasoning, LIMO, s1, etc.).
>
> We hope our additional experiments could resolve your concern.
>
>
> Best,
>
> The Authors

---

### Comment · Area_Chair_zV6Q · 2025-08-04
**Gentle Reminder: Please Reply to Authors’ Responses (Only if Not Yet Done)**

Dear Reviewers,

As the discussion deadline approaches, may we kindly ask you to review the authors’ responses and post a constructive reply—unless you have already done so, in which case please kindly disregard this gentle reminder.

Your thoughtful engagement is deeply appreciated and essential to a fair and timely process. With sincere thanks for your continued dedication.

Area Chair

---

### Note · Authors · 2025-08-13

We sincerely appreciate the AC and reviewers for their time, feedback, and constructive suggestions. We are delighted that the reviewers find our method to be effective (`vDcm`, `dezw`), our motivation to be strong (`MqSo`), our analysis to be comprehensive (`MqSo`), and our writing to be clear (`dezw`).

About the Generalizability and Novelty:
## Generalizability
+ **We have added results of LiveCodeBench, Business, and Law from MMLU-pro**. **DoubleChecker improves performance on these tasks, even though it was not trained on any related examples**.
+ **We have clarified how research papers of domain-specific tasks (math, code, or other topics) evaluate their methods** ( with `vDcm`). For example, prior works like DeepMath, DeepScaleR, and OpenMathReasoning focus solely on mathematical reasoning, while DeepCoder and OpenCodeReasoning focus exclusively on coding tasks. Research papers are not technical reports of LLMs produced by AI/tech companies.
+ **The data curation pipeline and training paradigm of DoubleChecker are directly applicable to other domains.**

## Novelty
+ **The scope is different**: Previous works usually focus on prompting (self-refine, reflextion) for fast-thinking LLMs (Learning to Check, S3CMath). There is no further self-reflection work for these slow-thinking LLMs, which are believed to have reflection-like reasoning. Our starting point is that we find "Aha Moment Does Not Equate to Effective Self-Critique" (in Section 3.1), and that is why we use specialized SFT to teach slow-thinking LLMs to self-critique.
+ **The specific design is different**: Compared with previous works that use self-generative feedback, our critique explicitly gives a final judgment (True/False) and uses this signal as the stop criterion of the self-improvement loop. Our SFT data curation pipeline is also different from traditional SFT.
+ **About the effectiveness**: A concurrent work (ThinkTwice) pioneers in prompting slow-thinking LLMs to revise previous solutions. Their preliminary results of SFT show no improvements compared with the prompting method, even with 100K training data. In contrast, our DoubleChecker can substantially improve the performance of original slow-thinking LLMs using only around 1.7K data.
+ **We also added a comparison to previous self-refine baselines**, and our DoubleChecker gains better results.

Thank you again for your efforts! We hope our further explanations will address the remaining concerns.

---

### Decision · Program_Chairs · 2025-09-17

**Decision:**

Reject

**Comment:**

Summary of the paper: This paper claims that long-CoT models exhibit only superficial “aha-moments” and cannot produce genuinely useful self-critiques that lead to systematic refinement. They demonstrate this empirically by showing that DeepSeek-R1 and similar distilled models rarely correct their own errors when left to reason autonomously. To remedy the limitation, the authors introduce Double-Checker, a training-plus-inference framework in which a model is first SFT on a carefully curated set of 1,730 instances that explicitly contain solution, critique, and revised-solution triplets, and then at test time is allowed to iterate the “critique → refine” loop until its internal self-critique signals satisfaction. Experiments on AIME-2024/25, MATH500, and GPQA reveal substantial absolute gains over strong long-CoT baselines (e.g., +18.2 % on AIME25 for a 32 B model), while ablations show that both the data format and the iterative procedure contribute to the improvement.

Strengths of the paper:
1. Empirical impact: Consistent, large accuracy boosts on challenging math and science benchmarks. Separate contributions of data format, critique data, and iterative rounds are quantified.
2. Data economy: Only 1,730 high-quality examples suffice to teach the self-critique skill, underscoring the power of targeted data curation.
3. Clear problem framing: Pinpoints the gap between apparent reflection and effective self-correction in long-CoT models.
4. Reproducible pipeline: Well-documented prompts and dataset construction steps facilitate follow-up work.

Weaknesses of the paper: After reading the rebuttal, it seems to me that it has largely satisfied the two reviewers. I would encourage the authors to add the additional experiments—particularly those that venture beyond math and reasoning domains—in the next revision. Reviewer dezw, however, remains unconvinced about the paper’s novelty. Self-reflection and fine-tuning are already well-explored, and the reliance on human-engineered priors for self-correction may curtail the method’s broader applicability.

Reasons for the decision: Despite the conceptual overlap with prior self-reflection work, this paper delivers a carefully engineered instantiation that yields unprecedented gains on widely used reasoning benchmarks. The evidence that a tiny, curated dataset can elicit such improvements is scientifically valuable, and the ablation studies provide useful insight into why it succeeds. While reviewer concerns about teacher-model dependence and domain breadth are valid, the authors provide new results of LiveCodeBench, Business, and Law from MMLU-pro. The proposed method improves performance on these tasks, even though it was not trained on any related examples.  and do not outweigh the demonstrated, reproducible advance in model reasoning performance. While my own assessment leans toward acceptance, I must respect the stringent resource constraints and exceptionally high bar set by NeurIPS 2025; under these circumstances, I regretfully find myself aligned with a decision to decline.